# Quantum dynamical effects of vibrational strong coupling in chemical reactivity

Lachlan P. Lindoy [1], Arkajit Mandal [1] & David R. Reichman [1] ✉

Recent experiments suggest that ground state chemical reactivity can be modified when placing molecular systems inside infrared cavities where molecular vibrations are strongly coupled to electromagnetic radiation. This phenomenon lacks a firm theoretical explanation. Here, we employ an exact quantum dynamics approach to investigate a model of cavity-modified chemical reactions in the condensed phase. The model contains the coupling of the reaction coordinate to a generic solvent, cavity coupling to either the reaction coordinate or a non-reactive mode, and the coupling of the cavity to lossy modes. Thus, many of the most important features needed for realistic modeling of the cavity modification of chemical reactions are included. We find that when a molecule is coupled to an optical cavity it is essential to treat the problem quantum mechanically to obtain a quantitative account of alterations to reactivity. We find sizable and sharp changes in the rate constant that are associated with quantum mechanical state splittings and resonances. The features that emerge from our simulations are closer to those observed in experiments than are previous calculations, even for realistically small values of coupling and cavity loss. This work highlights the importance of a fully quantum treatment of vibrational polariton chemistry.

A series of recent experiments[1–12] have suggested that when molecular vibrations are coupled to the radiation modes inside an optical cavity, ground state chemical kinetics can be both enhanced[1,2,8] or suppressed[3,6,8]. Such effects are purported to operate in the absence of external optical pumping[13] and have been hypothesized to arise from the hybridization of molecular vibrational states and the photon (Fock) states of cavity radiation mode[1,2]. The interpretation of these experiments is still a matter of debate, and thus the viability of marked alterations in chemical reactivity remains an open topic. Indeed, while the spectroscopic fingerprints of light–matter hybridization, such as the Rabi-splitting observed in the IR spectra are manifest, the significance of the coupling to radiation modes for markedly changing chemical reactivity is unclear.

Theoretical studies that describe both the radiation modes as well as the molecular vibrations using classical mechanics have successfully revealed alterations in reactivity but have had limited success in describing currently available experiments[14–22]. Specifically, previous work[14–16] using the Grote–Hynes (GH) rate theory[23–25], applicable in the limit of strong molecule-bath interactions, do show cavity frequency-dependent chemical kinetics modification. However, these studies[14–16] predict that the chemical reaction rate is suppressed most strongly when the cavity frequency is near to the barrier frequency as opposed to molecular vibrational frequencies as seen in experiments, and that the rate profile is only weakly modified in an extremely broad manner with respect to the cavity frequency $\omega_c$, spanning thousands of wavenumbers (~5000 cm$^{-1}$). This is in stark contrast to experimental observations, where the width of the changes in the rate profile are on the order of ~100 cm$^{-11-6}$. On the other hand, a recent study[17] using the Pollak-Grabert-Hänggi (PGH) theory[26], as well as direct trajectory-based computational work[20,27], have predicted enhancement of chemical rates when the molecule-bath coupling is relatively weak. Interestingly, these studies have predicted a significantly sharper rate profile than that which emerges at strong coupling, have demonstrated that the effect is more sizable, and have revealed that the chemical rate is most strongly enhanced when the cavity frequency is close to the reactant vibrational frequency[17,20,27]. While these studies move theory closer to laboratory

[1]Department of Chemistry, Columbia University, 3000 Broadway, New York, NY 10027, USA. ✉e-mail: drr2103@columbia.edu

observations, there is still a substantial gulf between experiments and our theoretical understanding.

A major missing component in the theoretical work discussed above is the quantum nature of the problem. Classical treatments of chemical kinetics are often capable of capturing room temperature reaction rates to within an order of magnitude compared to the exact quantum mechanical ones in model calculations[28]. However, as revealed in this work, in comparison to the exact quantum calculations, the classical approaches do not capture the characteristic features of the cavity-modified chemical reactivity (namely, the sharp resonant enhancement and suppression of chemical rate) seen in experimental work[1–6].

Simple, approximate quantum corrections to the GH theory, such as found using quantum transition state theory[29] or zero-point energy corrections to the energy barrier[30] have been carried out, but these approximate calculations diverge from experimental expectations even more than their fully classical counterparts, showing, for example, an even broader range of alteration of the rate profile than that seen in classical calculations[29]. Recent fully quantum dynamical studies which ignore the explicit interactions of the molecule with the solvent degrees of freedom also do not find a resonant structure in the cavity frequency dependence of chemical rate[30,31]. Taken as a whole, these studies point to the clear pressing need to perform exact quantum calculations on models that include the relevant molecular, solvent, and cavity degrees of freedom.

We note that in essentially all previous experimental work[1–8], a large ensemble of molecules (~$10^{10}$ molecules per cavity mode[32,33]) are collectively coupled to the cavity radiation modes. In contrast, most theoretical work, including the calculations we present here, operates in the single molecule limit and does not address collective effects in a direct manner. Despite some studies addressing collective polaritonic behavior[16,34,35], a detailed theoretical explanation for such collective effects remain elusive. To this end, the work presented here does not directly address the issue of collective effects. We would like to point out, however, that some idealized models of collective polaritonic behavior[16] reduce to models similar to those considered below (namely, a reaction coordinate strongly coupled to a spectator/collective solvent mode). In this regard, as will be discussed below, it is crucial to note that our quantum calculations suggest that alterations to reaction rates may be observed with cavity coupling strengths that are orders of magnitude smaller than suggested in other recent single molecule studies.

In this work, we use a customized version of the hierarchical equations of motion (HEOM) approach[36] (see Methods for further details) to exactly simulate the cavity-modified chemical kinetics of a single molecule coupled to a radiation mode as well as dissipative molecular and solvent modes. We emphasize that compared with previous work[30,31], which employs a quantum treatment of the cavity and the molecular degrees of freedom in the absence of a molecular or solvent bath, we explicitly include the dissipation provided by a solvent exactly within our model. As we will discuss, this is crucial for obtaining the sharp, resonant cavity modifications observed here. We show that coupling molecular vibrations to a cavity radiation mode can both enhance or suppress chemical reactivity, with the largest effect occurring when the cavity mode is near-resonant with specific molecular vibrational modes. Crucially, we find that the cavity frequency-dependent rate shows a much sharper profile (~100 cm$^{-1}$) compared to what is predicted from classical rate theories[14,17] or with quantum corrections[21,30], even when compared to recent theoretical results in the classical weak-coupling regime[20,27]. Our results also demonstrate that the details of the solvent-molecule interactions are extremely important, and that even realistically small rates of cavity loss can play a crucial role in enabling cavity modification of chemical reactivity. Overall, our results reveal that the cavity modification of chemical rates can largely be rationalized by considering how the molecular vibrational states are altered by hybridization with the cavity photon

states (forming so-called vibrational polaritons) to effectively increase or decrease the interaction of the molecule with its environment. The resonant structure in the cavity-modified reaction rate naturally arises from the hybridization of light and matter, which occurs most strongly when the cavity and matter states are in resonance. These effects emerge from fundamental quantum light–matter interactions and cannot be fully captured with simple classical or semiclassical descriptions of light and matter.

## Results

### Theoretical model

In this work, we have systematically explored a wide range of parameter regimes associated with cavity modification of chemical reactivity in the single molecule limit. There are three parts to our model system:

- The molecular subsystem, which is described by a double well potential profile which is widely employed in studying chemical kinetics[24,28,37–39] and used previously in the theoretical investigation of vibrational polariton chemistry[14–16,29–31,40]. We find that, outside of the tunneling-dominated regime (see Supplementary Note 3) the rough qualitative features of our results do not sensitively depend on the parameters describing the molecular system, such as the well/barrier frequency, height of the barrier, driving force and or shape of the solute dipole (either linear or nonlinear).
- The cavity radiation and light–matter interaction term. Here, there are three relevant parameters which characterize this part of our model, namely the cavity photon frequency, light–matter coupling strength, and cavity lifetime. We find that all three parameters play a crucial role in modifying chemical reactivity.
- The solvent and the molecule–solvent interactions. We find that the spectral density that characterizes the molecule–solvent interactions plays a crucial role in the cavity modification of chemical reactivity.

We note that while it appears that there are a large number of adjustable parameters, we have, where possible, aimed to either scan over parameter space or use physically relevant parameters.

The model quantum electrodynamics (QED) Hamiltonian used in this work is based on the Pauli–Fierz (PF) light–matter Hamiltonian in the dipole gauge in the single mode and long-wavelength limits, and is written as[41–43]

$$\hat{H} = \hat{H}_{\mathrm{mol}} + \hat{H}_{\mathrm{solv}} + \hat{H}_{\mathrm{cav}} + \hat{H}_{\mathrm{loss}}, \tag{1}$$

where $\hat{H}_{\mathrm{mol}}$ is the molecular Hamiltonian, $\hat{H}_{\mathrm{solv}}$ describes solvent as well as molecule–solvent interactions, $\hat{H}_{\mathrm{cav}}$ is the cavity Hamiltonian describing a radiation mode and its interaction to matter in the dipole gauge, and $\hat{H}_{\mathrm{loss}}$ describes the cavity loss term.

Before discussing the details of the light–matter interactions, we first present the model used for our matter degrees of freedom. Here we make use of a Caldeira–Leggett-like model[44] in which the molecular degrees of freedom are bilinearly coupled to a harmonic bath (solvent). In this work we consider a molecular Hamiltonian $\hat{H}_{\mathrm{mol}} = \hat{T}_R + V(\hat{R})$ that contains a one-dimensional reaction coordinate $R$. The ground state potential energy surface along this reaction coordinate, $V(\hat{R}) = \frac{\omega_b^4}{16E_b} \cdot \hat{R}^4 - \frac{1}{2}\omega_b^2 \cdot \hat{R}^2 - c \cdot R^3$, takes the form of double well potential. In the main text we consider a barrier frequency $\omega_b = 1000$ cm$^{-1}$, barrier $E_b = 2250$ cm$^{-1}$, and a symmetric double potential with $c = 0$ (see Supplementary Note 3 for results with $c \neq 0$ and for other values for the barrier height and frequency), as shown in Fig. 1a (black solid line). The molecular Hamiltonian $\hat{H}_{\mathrm{mol}}$ can be equivalently represented using

the vibrational states,

$$
\begin{aligned}
\hat{H}_{\mathrm{mol}} = \sum_i E_i |v_i\rangle\langle v_i| &\equiv \bar{E}_0 \left( |v_R\rangle\langle v_R| + |v_L\rangle\langle v_L| \right) \\
&+ \sum_{i \geq 2} E_i |v_i\rangle\langle v_i| + \Delta \left( |v_R\rangle\langle v_L| + |v_L\rangle\langle v_R| \right),
\end{aligned}
\tag{2}
$$

where $\{|v_i\rangle\}$ are the vibrational eigenstates of the molecular Hamiltonian ($\hat{H}_{\mathrm{mol}}|v_i\rangle = E_i|v_i\rangle$). In the second line we have introduced localized states $|v_L\rangle = \frac{1}{\sqrt{2}}(|v_0\rangle + |v_1\rangle)$ and $|v_R\rangle = \frac{1}{\sqrt{2}}(|v_0\rangle - |v_1\rangle)$, with an energy $\bar{E}_0 = \frac{1}{2}(E_0 + E_1)$ and a coupling $\Delta = \frac{1}{2}(E_1 - E_0)$. These states are the localized ground states of the left and the right wells (blue and red wavefunctions in Fig. 1a), respectively. We define the well frequency $\omega_0 = E_2 - \bar{E}_0 \approx 1140\ \mathrm{cm}^{-1}$.

The solvent contribution to the Hamiltonian $\hat{H}_{\mathrm{solv}}$ is taken as

$$
\begin{aligned}
\hat{H}_{\mathrm{solv}} = &\frac{\hat{P}_Q^2}{2} + \frac{1}{2}\omega_Q^2 \left( \hat{Q} + \frac{C_Q \hat{R}}{\omega_Q^2} \right)^2 \\
&+ \sum_j \frac{\hat{P}_j^2}{2} + \frac{1}{2}\Omega_j^2 \left( \hat{X}_j + \frac{C_j \hat{R}}{\Omega_j^2} \right)^2 \\
&+ \sum_j \frac{\hat{p}_j^2}{2} + \frac{1}{2}\omega_j^2 \left( \hat{x}_j + \frac{c_j \hat{Q}}{\omega_j^2} \right)^2 .
\end{aligned}
\tag{3}
$$

The first line describes a spectator mode (e.g., an intramolecular vibrational mode that is orthogonal to the reaction coordinate, or

equivalently a collective solvent mode[45,46] as has been considered in simplified models of collective VSC[16]) with coordinate $Q$ coupled to the reaction coordinate. The second line describes a set of dissipative solvent modes described by a broad spectral density $J_U(\Omega) = \frac{\pi}{2}\sum_j \frac{C_j^2}{\Omega_j}\delta(\Omega - \Omega_j) = 2\Lambda_s \Omega\Gamma/(\Omega^2 + \Gamma^2) = \eta_s \Omega\Gamma^2/(\Omega^2 + \Gamma^2)$ (black dashed line in Fig. 1a). Here $\Lambda_s$ corresponds to the solvent reorganization energy, $\Omega$ is the characteristic frequency that determines the peak of the spectral density, and $\eta_s = 2\Lambda_s$ is the friction constant. The third line describes a set of secondary solvent modes $\hat{x}_j$ that couple to the spectator mode coordinate $Q$ and are also described with a broad spectral density $J_u(\omega) = \frac{\pi}{2}\sum_j \frac{c_j^2}{\omega_j}\delta(\omega - \omega_j) = 2\lambda_s\omega\gamma/(\omega^2 + \gamma^2)$. Here, $\lambda_s$ and $\gamma$ are the reorganization energy and the characteristic frequency of this secondary solvent bath, respectively.

In this work, the essential features of the solvent-molecule interactions are included in the spectral density. In particular, the spectral density can be calculated by molecular dynamics in simulations and often contains sharp peaks[47–51]. When this is the case, the spectral density can roughly be grouped into two categories. In the first category are cases where the spectral density has peaks near the molecular vibrational frequencies. The second category comprises cases where the spectral density does not have peaks near the molecular vibrational frequencies. We find that spectral densities with off-resonant peaks exhibit the same cavity modulation of reactivity as spectral densities devoid of peaks, as shown in Supplementary Note 8.

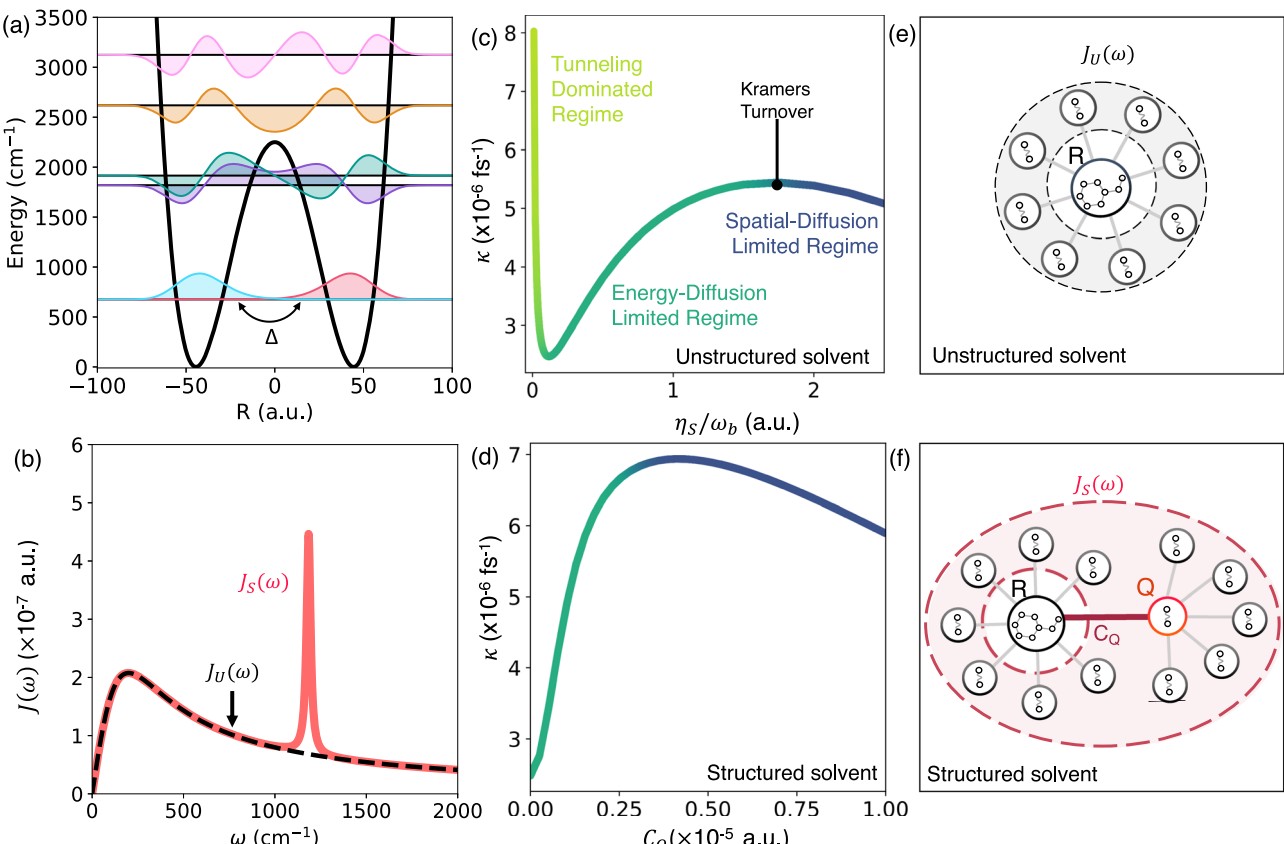

**Fig. 1 | Effect of molecule–solvent coupling on chemical kinetics. a** Potential energy surface along the reaction coordinate $R$ and vibrational eigenstates of the model molecular system. The localized ground state of the left and the right well is coupled through $\Delta$, which induces tunneling. **b** Effective spectral density for unstructured ($J_U(\omega)$, black dashed line) and structured ($J_S(\omega)$, red solid line) solvent environment as schematically depicted in (**e**, **f**), respectively. The peak at 1200 cm⁻¹ in $J_S(\omega)$ arises due to strong coupling ($C_Q$) between the molecule and a spectator mode $Q$. **c** Chemical rate constant $\kappa$ as a function of solvent friction $\eta_s$ when the molecule is embedded in an unstructured environment, as illustrated in (**e**). **d** Chemical rate constant $\kappa$ as a function of molecule-spectator mode coupling $C_Q$ when the molecule is embedded in a structured environment as illustrated in (**f**). Source data are provided as a Source Data file.

This is why in the following simulations, we will consider two specific cases for the solvent. In the first, we will consider no coupling between the system and spectator mode, i.e., $C_Q = 0$. In this case, the spectral density for the solvent is described by the broad, unstructured spectral density $J_L(\omega)$, and as such, will be referred to as the "unstructured" environment. In the second, we will consider the non-zero coupling between the system and spectator mode. In this case, applying a normal mode transformation to the solvent Hamiltonian, we find that the presence of the spectator mode gives rise to a sharp peak in the solvent spectral density $J_S(\omega)$, as shown in Fig. 1b). Note that here we use the term "structured" to refer to the sharp structure in the spectral density at specified frequencies which does not imply positional or orientational structure in the environment.

This model of a bilinearly coupled harmonic solvent is known to capture solvent-mediated dissipative processes[52] and is central to many rate theories in the liquid-state[26,44,53–55] as well as numerical treatments of chemical reaction processes that include dissipative effects due to condensed phase environmental degrees of freedom[37,56]. For regimes in which a linear response treatment of the solvent degrees of freedom is valid, an anharmonic solvent can rigorously be mapped onto such harmonic bath model[57–59].

The cavity Hamiltonian $\hat{H}_{cav}$ describing a single-cavity mode and its coupling to matter is given by

$$\hat{H}_{cav} = \frac{\hat{p}_c^2}{2} + \frac{1}{2}\omega_c^2\left(\hat{q}_c + \sqrt{\frac{2}{\omega_c}}\boldsymbol{\eta_c}\cdot\hat{\boldsymbol{\mu}}\right)^2. \tag{4}$$

Here, $\omega_c$ is the cavity photon frequency and $\boldsymbol{\eta_c} = \eta_c\hat{\boldsymbol{e}} = \frac{1}{\omega_c}\sqrt{\frac{\hbar\omega_c}{2\epsilon_0 V}}\hat{\boldsymbol{e}}$ is the light−matter coupling vector with vacuum permittivity $\epsilon_0$, quantization volume $V$, the direction of polarization $\hat{e}$ and light−matter coupling strength $\eta_c$. Further, $\hat{p}_c = i\sqrt{\frac{\hbar\omega_c}{2}}(\hat{a}^\dagger - \hat{a})$ and $\hat{q}_c = \sqrt{\frac{\hbar}{2\omega_c}}(\hat{a}^\dagger + \hat{a})$, where $\hat{a}^\dagger$ and $\hat{a}$ are the photon creation and annihilation operators, and $\hat{\boldsymbol{\mu}}$ is the matter dipole operator vector. In all calculations, we will take the matter dipole operator vector to point along the direction of polarization, that is $\hat{\boldsymbol{\mu}} = \hat{\mu}\hat{\boldsymbol{e}}$.

Finally $\hat{H}_{loss}$ describes the bath that is coupled to the cavity mode, which enables cavity loss,

$$\hat{H}_{loss} = \sum_k \frac{\hat{\Pi}_k^2}{2} + \frac{1}{2}\tilde{\omega}_k^2\left(\hat{Q}_k + \frac{\mathcal{C}_k\hat{q}_c}{\tilde{\omega}_k^2}\right)^2, \tag{5}$$

where $\tilde{\omega}_k$ and $\mathcal{C}_k$ which control cavity leakage are described via spectral density $J_L(\omega) = \frac{\pi}{2}\sum_k \frac{\mathcal{C}_k^2}{\tilde{\omega}_k}\delta(\omega - \tilde{\omega}_k) = 2\lambda_L\omega\gamma_L/(\omega^2 + \gamma_L^2)$. Here, $\lambda_L$ and $\gamma_L$ are the reorganization energy and the characteristic frequency of the far-field modes that dissipate the cavity radiation mode. With this spectral density, the cavity loss rate is defined as $\Gamma_c = 1/\tau_c = 2J(\omega_c)/(\omega_c(1 - e^{-\beta\omega_c}))$ where $\tau_c$ is the cavity lifetime and $\beta = 1/k_BT$ with the Boltzmann constant $k_B$ and temperature $T$, in all simulations, a temperature of $T = 300$ K was considered. Physically, our model Hamiltonian flexibly contains nearly all essential ingredients assumed to influence chemical reactions in a cavity. Further details of the model parameters are provided in Supplementary Note 1.

### Chemical kinetics outside a cavity

The chemical kinetics of a molecular system in the absence of the cavity (setting $\eta_c = 0$) embedded in an unstructured solvent (setting $C_Q = 0$) depends on the molecule−solvent interaction strength. Such a setup is schematically illustrated in Fig. 1e and the corresponding bath spectral density $J_U(\omega)$ is presented in Fig. 1b (black dashed line). The chemical rate constant $\kappa$ (see Methods), obtained from exact quantum dynamics simulation using a specialized HEOM approach (see

Methods) as a function of $\eta_s$ is presented in Fig. 1c and shows three distinct regimes. For very low $\eta_s$ the chemical kinetics is dominated by direct nuclear tunneling, that is, the transition $|v_L\rangle \to |v_R\rangle$ via the tunnel-coupling $\Delta$ in Eq. (2). We refer to this regime as the tunneling-dominated regime. In this regime, an increase in the molecule−solvent interaction (by increasing $\eta_s$) leads to a sharp decline in the reaction rate[60] as the bath (solvent) degrees of freedom effectively renormalize and lower $\Delta$.

While the increase in $\eta_s$ reduces the direct nuclear tunneling, an alternate reaction pathway involving thermal excitations, that is $|v_L\rangle \to \{|v_i\rangle\} \to |v_R\rangle$, starts to play an increasingly important role. For $\eta_s > 0.1\omega_b$, the later pathway becomes the dominant one. For $1.8\omega_b > \eta_s > 0.1\omega_b$, the overall reaction rate is limited by the equilibration rate of the vibrational states in the left well. In this regime, akin to the energy diffusion-limited regime in the Kramers turnover problem[24], the overall reaction rate increases with increasing $\eta_s$ as can be seen in Fig. 1c. Note that throughout this work we will refer to the peak in rate vs coupling plot as the Kramers turnover point, and, interchangeably, the regime before the peak as the weak-coupling or "energy diffusion-limited" regime and that after the peak as the strong-coupling or "spatial diffusion-limited" regime. It should be noted that when the reaction rate is controlled by multiple parameters, the use of the terms such as "energy diffusion-limited" or "spatial diffusion-limited" may be an imprecise means to distinguish the pre- and post-turnover regimes[26]. Finally, further increases in $\eta_s$ drives the system into the spatial diffusion-limited regime where the reaction rate decreases with increasing solvent friction $\eta_s$. In classical rate theory, this transition from the energy diffusion-limited regime to the spatial diffusion-limited regime is referred to as the Kramers turnover[24].

For a molecular system embedded in a structured solvent outside of the cavity ($\eta_c = 0$ in $\hat{H}$), the chemical rate as a function of the molecule-spectator mode coupling $C_Q$ is presented in Fig. 1d. The corresponding molecular system is also schematically illustrated in Fig. 1f. The effective bath spectral density[45,46] $J_S(\omega) = \frac{\pi}{2}\sum_j \frac{\tilde{C}_j^2}{\tilde{\omega}_j}\delta(\omega - \tilde{\omega}_j)$, for the effective (and equivalent) solvent Hamiltonian $\hat{H}_{solv} \equiv \sum_j \frac{\tilde{p}_j^2}{2} + \frac{1}{2}\tilde{\omega}_j^2(\tilde{x}_j + \tilde{C}_j\tilde{Q}/\tilde{\omega}_j^2)^2$, is presented in Fig. 1b (red solid line). In comparison to the unstructured environment, the spectral density $J_S(\omega)$ for the structured environment shows a sharp spike at 1200 cm$^{-1}$, which originates from the spectator mode $Q$ with frequency $\omega_Q = 1200$ cm$^{-1}$, and the width of this peak originates from the secondary solvent spectral density $J_u(\omega)$. This is characteristic of complex molecular environment[61–63] where the spectral density contains numerous spikes. It is worth noting that the reaction coordinate is strongly coupled to other orthogonal vibrations[64] in molecular systems considered in typical experimental work[2].

Here we will consider $\eta_s = 0.1\omega_b$, so that the system is in the weak-coupling (pre-turnover) regime for $C_Q = 0$. Similar to Fig. 1c, in Fig. 1d, the reaction rate initially increases as $C_Q$ increases, and undergoes a turnover, at $C_Q \approx 3 \times 10^{-6}$ a.u., following which the rate decreases with increasing $C_Q$, akin to the Kramers turnover.

### Chemical kinetics inside a cavity

When the molecular system is placed inside an optical cavity (schematically illustrated in Fig. 2a), light−matter coupling between molecular vibrational states leads to the formation of vibrational polaritons. This is shown in Fig. 2b, c. Here, we consider first a molecular system embedded in an unstructured environment (as depicted in Fig. 1e) with the reaction coordinate directly coupled to the cavity radiation mode (with $\hat{\mu} = \hat{R}$). In Supplementary Note 6, we consider a nonlinear dipole operator and find that the general conclusions drawn in this section do not sensitively depend on the exact form of the dipole moment operator.

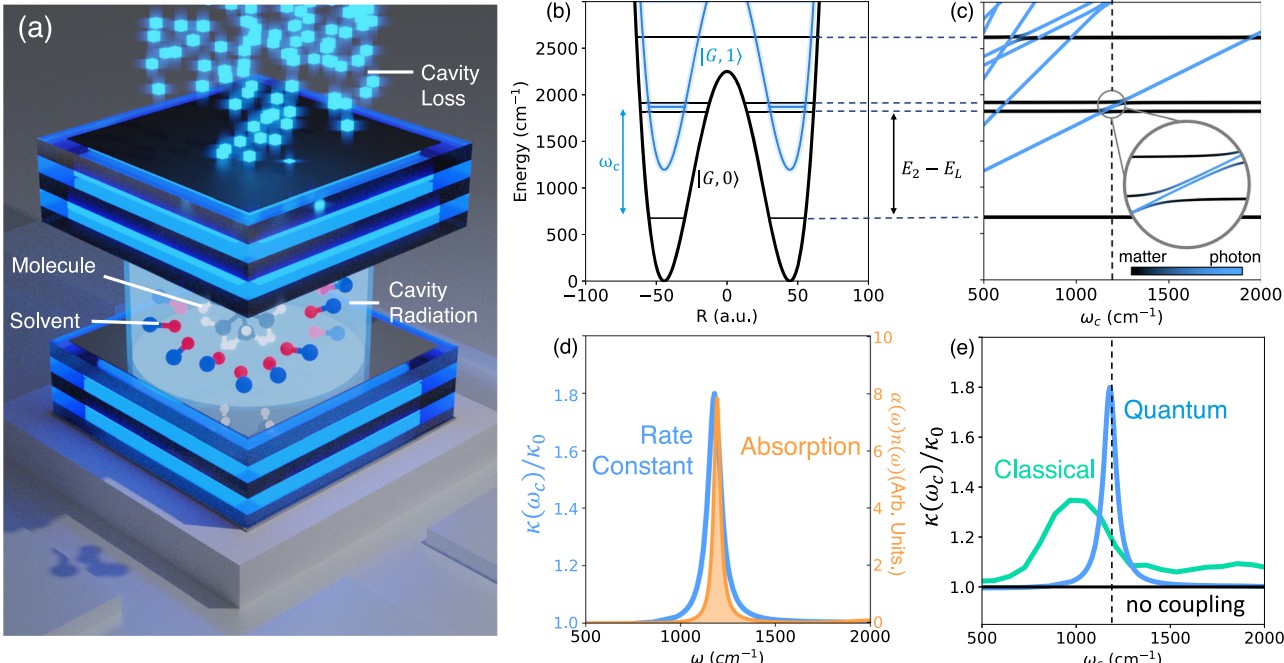

**Fig. 2 | Resonant cavity modification of ground state chemical kinetics. a** Schematic illustration of a molecule coupled to a lossy cavity radiation mode as well as to other solvent molecules. **b** Potential energy surfaces as a function of the reaction coordinate $R$ for a molecular ground adiabatic state with 0 photons $|G,0\rangle$ (black solid line) and with 1 photon (blue solid line) $|G,1\rangle$, as well as the corresponding vibrational eigenstates of $|G,0\rangle$ and $|G,1\rangle$ (horizontal solid lines). Here, $E_L = E_R$ are the energy of the localized ground state of the left and the right well and $E_2$ is the energy of the first vibrational excited state of the bare molecule. **c** Vibrational polariton eigenspectrum as a function of cavity photon frequency $\omega_c$. **d** Absorption spectrum (yellow shaded area) as a function of frequency $\omega$ of the

system interacting with the solvent uncoupled from the cavity and the cavity-modified rate constant $\kappa$ (normalized with the rate constant $\kappa_0$ outside of the cavity) as a function of photon frequency $\omega = \omega_c$ with a cavity lifetime $\tau_c = 100$ fs (blue solid line). The linewidth of the absorption spectrum and chemical rate constant are similar in magnitude. **e** Comparing the cavity-modified chemical rate constant computed using exact quantum (blue solid line) and classical (green solid line) dynamical simulations, showing the failure of classical description in quantitatively capturing the effects of quantum light–matter interactions. Here, the light–matter coupling is $\eta_c = 0.00125$ and the solvent friction is $\eta_s = 0.1\omega_b$. Source data are provided as a Source Data file.

In the simplified Jaynes–Cummings model[65], excited vibrational states (of the ground electronic state $|G\rangle$) with 0 photons in the cavity $|v_2\rangle \otimes |G,0\rangle \equiv |v_2\rangle \otimes |0\rangle$ hybridize with the ground vibrational states with 1 photon in the cavity, $|v_L\rangle \otimes |1\rangle$, through the coupling $\langle v_2, 0|\hat{H}_{cav}|v_L,1\rangle = \eta_c\omega_c\langle v_L|\hat{R}|v_2\rangle$. This results in an avoided crossing of polaritonic eigenenergies when $\omega_c \approx \omega_0 = E_2 - \bar{E}_0 \approx 1140$ cm$^{-1}$ in Fig. 2c, which results in a Rabi-splitting $\Omega_c$ (~30 cm$^{-1}$ for $\eta_c = 0.00125$ used in Fig. 2), which can be observed in absorption spectra[1–4,6]. The normalized light–matter coupling $\bar{\eta} = \frac{\Omega_c}{2\omega_c}$, which places the light–matter coupling in the weak-strong ($\bar{\eta} < 0.1$), ultra-strong ($1.0 > \bar{\eta} > 0.1$) and deep-strong ($\bar{\eta} > 1$) coupling regimes, is approximately related to $\eta_c$ as $\bar{\eta} \approx \eta_c/\sqrt{2\omega_0}$ when approximating the reactant well as harmonic and using the Jaynes–Cummings model. All results presented here lie in the weak-strong coupling regime and the results presented in Fig. 2 use $\bar{\eta} \approx 0.0125$.

Using our HEOM approach, we have simulated the chemical kinetics of this molecule-cavity hybrid system. The resultant chemical rate profile as a function of the cavity frequency $\omega_c$ is found to be sharply peaked around 1200 cm$^{-1}$, the frequency at which the cavity is resonant with the molecular vibrational transitions. Remarkably, the chemical rate profile (blue solid line in Fig. 2d) has a similar lineshape compared to the absorption spectra (yellow shaded area in Fig. 2d). This bears a striking resemblance to recent experimental observations[1–8]. We must emphasize, that current experiments operate in a collective regime, while the present theoretical calculations pertain to a single-molecule, single-cavity setup. Notably, however, the light–matter couplings used in this work are much smaller than what has been used in recent theoretical work in the single molecule case[14,15,20,22,27,31,66]. For example, a quantization volume of 0.19 nm$^3$ was used in ref. 66 while we use $V$ ~10–288 nm$^3$ here (see Supplementary Note 10).

The photon frequency-dependent rate profile obtained with exact quantum dynamics is much sharper, with a full width at half maximum (FWHM) ~85 cm$^{-1}$, than the rate profile obtained using direct classical simulations, FWHM of ~400 cm$^{-1}$ (using generalized Langevin equation, see Supplementary Note 4), as shown in Fig. 2e. Additionally, the classical results show a significantly smaller peak enhancement that occurs at a lower frequency (~1000 cm$^{-1}$) than in our exact quantum calculation (which has a peak at ~1190 cm$^{-1}$, compared to the peak in the IR absorption spectrum at 1200 cm$^{-1}$). Similar detunings (in the range of ~150–200 cm$^{-1}$) of the peak in the classical rate profile compared to the vibrational peaks of the molecular system have been observed in prior classical rate theory calculations[17], and in molecular dynamics simulations[20]. In Supplementary Note 4, we compare the classical rate profile to the IR spectrum obtained using classical methods and observe discrepancies between the peak locations of the two profiles. Interestingly, the width of the classical absorption profile is as sharp as its quantum counterpart shown in Fig. 2d (also see Fig. S6) in contrast to the width of the chemical rate profile obtained classically. This difference is due to the fact that the IR spectrum, which depends on molecule–solvent interactions, is accurately described within the Golden Rule formalism, while the chemical kinetics inside the cavity, where resonance effects are important, is not. These results underscore the importance of the quantum mechanical interplay of photonic and molecular degrees of freedom in modifying chemical reactivity.

In Fig. 3, we analyze the role of light–matter coupling strength, cavity loss, and molecule–solvent coupling strength to gain mechanistic insights into cavity-modified chemical kinetics. Figure 3a presents the normalized reaction rate constant $\kappa(\omega_c)/\kappa_0$ (with $\kappa_0$ the rate constant outside cavity) at various light–matter coupling strengths $\eta_c$. Increasing the light–matter coupling increases both

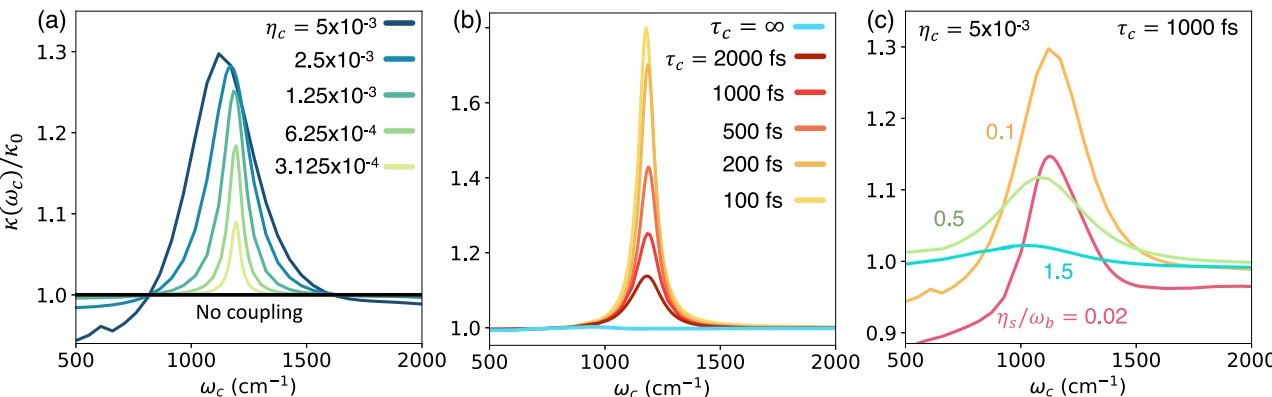

**Fig. 3 | Effect of various parameters on cavity-modified chemical reactivity.**
**a** Effect of light–matter coupling $\eta_c$ on chemical reactivity with cavity lifetime $\tau_c = 1000$ fs and solvent friction $\eta_s = 0.1\omega_b$ where $\omega_b$ is the barrier frequency. Here, $\kappa(\omega_c)$ is the cavity-modified rate constant as a function of the photon frequency $\omega_c$ and $\kappa_0$ is the rate constant outside the cavity. **b** Effect of cavity lifetime $\tau_c$ at the width as well as the height of the rate profile. This is expected, as $\eta_c = 0.00125$ and $\eta_s = 0.1\omega_b$. Note the negligible cavity modifications in the absence of cavity loss $\tau_c = \infty$. **c** Effect of solvent friction on chemical reactivity with cavity lifetime $\tau_c = 1000$ fs and light–matter coupling $\eta_c = 0.00125$. Source data are provided as a Source Data file.

the width as well as the height of the rate profile. This is expected, as higher coupling increases the effective environmental friction to the molecular system in the weak-coupling regime. At the same time, higher coupling also allows the cavity and vibrational excitation to hybridize at larger detuning, resulting in a wider rate profile.

Here, we also observe an off-resonant suppression of the chemical rate at higher light–matter couplings. This effect can be understood by applying a polaron transformation, leading to a rescaling of $\Delta \rightarrow \Delta \cdot \langle \exp[\pm i\eta_c \sqrt{2/\omega_c}\hat{p}_c \Delta\mu]\rangle$ due to the difference in permanent dipoles ($\Delta\mu = \langle v_L|R|v_L\rangle - \langle v_R|R|v_R\rangle$) between the right and left wells, resulting in a matter state-dependent displacement of the cavity mode $q_c$[67]. Similar effects have been investigated in the context of cavity-modified photo-dissociation[67] or cavity-mediated non-adiabatic electron transfer reactions[68,69].

Figure 3b presents the effects of cavity loss on the cavity-modified chemical kinetics. We observe that an increase in cavity loss leads to a significant increase in the reaction rate at resonance but leads to only minor changes off-resonance. The effect can be accounted for by considering the energy transfer processes occurring between the molecule-cavity subsystem (containing only $R$ and $q_c$ and described by $\hat{H}_{mol} + \hat{H}_{cav}$) when coupled to a dissipative bath composed of the solvent degrees of freedom $\{X_j\}$ and how the presence of far-field cavity modes $\{\mathcal{Q}_k\}$ modifies these processes. In the absence of cavity loss, thermalization of the cavity mode can only occur through energy transfer between the cavity mode and the molecular bath that are mediated by the reaction coordinate. Consequently, the cavity mode does not provide an efficient mechanism for energy loss, as excess energy transferred to the cavity mode during the reaction will necessarily transfer back to the reaction coordinate as the system approaches equilibrium. The inclusion of far-field modes that couple to the cavity mode in $\hat{H}_{loss}$ provide an additional pathway for the thermalization of the cavity mode that does not require energy transfer through the reaction coordinate. This allows for the cavity mode to act as an additional source of environmental friction for the reaction coordinate, which leads to the enhancement of the rate constant kinetics in the weak-coupling regime.

Consequently, we find that cavity modification of the reaction rate is negligible in the absence of cavity loss ($\tau_c \rightarrow \infty$, solid blue line) and shows no resonance structure (see further details in Supplementary Note 5). Therefore, cavity loss plays a significant role in modifying chemical reactivity in this regime.

This result also opens up interesting questions regarding the collective cavity modification of chemical reactivity. When considering

$N$ non-interacting molecules[19], from the perspective of one reactive molecule, the coupling of the rest of the molecules to the radiation mode is structurally the same as the far-field modes that couple to the cavity mode describing cavity loss. Therefore, the collective coupling may provide an additional source of dissipation (playing a similar role to cavity loss) and may modify chemical reactivity, but to what extent remains an open question reserved for future study.

In Supplementary Note 4, we present classical simulations for the situations considered in Fig. 3a, b. Here, we find that a purely classical description gives rise to qualitatively different conclusions. In the absence of cavity loss, the purely classical treatment predicts a resonant structure in the rate profile and shows minor sensitivity to the cavity loss.

In Fig. 3c, we investigate the role of the molecule–solvent interaction strength on the cavity-modified reaction rate at $\eta_c = 0.005$ and $\tau_c = 1000$ fs. For very weak coupling, $\eta_s = 0.02\omega_b$ a.u. (red solid line), we observe off-resonant suppression as well as resonant enhancement of the chemical reaction rate. Physically, this occurs since, for low $\eta_s$, both tunneling, as well as thermal activation (such as in the energy diffusion-limited regime) reaction pathways play important roles in the chemical reaction. The cavity suppresses the tunneling by renormalizing $\Delta$ as discussed above, while at the same time resonantly enhancing the thermal activation pathway by effectively increasing the environmental friction. The combination of these two competing effects leads to both off-resonant suppression and resonant enhancement of the rate at $\eta_s = 0.02\omega_b$. For $\eta_s = 0.1\omega_b$, as tunneling plays a less significant role, off-resonant suppression becomes less prominent. Thus at $\eta_s = 0.1\omega_b$, the cavity-modified rate constant primarily shows resonant enhancement. Further increase in $\eta_s$ decreases the extent of cavity modification as higher $\eta_s$ places the system closer to the turnover region. Thus, for $\eta_s = 1.5\omega_b$ (near the Kramers turnover, see Fig. 1c), the cavity modification becomes negligible. Due to the computational expense of our HEOM calculations, the simulation of the cavity-modified reaction rate is outside our reach for $\eta_s > 1.5\omega_b$. However, we speculate that cavity-mediated suppression could be observed at higher $\eta_s$. This will be explored using more efficient but approximate quantum dynamics approaches in future work.

So far, we have investigated a molecular system embedded in an unstructured environment. In Fig. 4, we consider the molecular system embedded in a structured environment, as depicted in Fig. 1f, characteristic of real molecular systems. Here we consider two scenarios, one where the cavity radiation mode is directly coupled to the reaction coordinate $R$, and one where it directly couples to the spectator mode

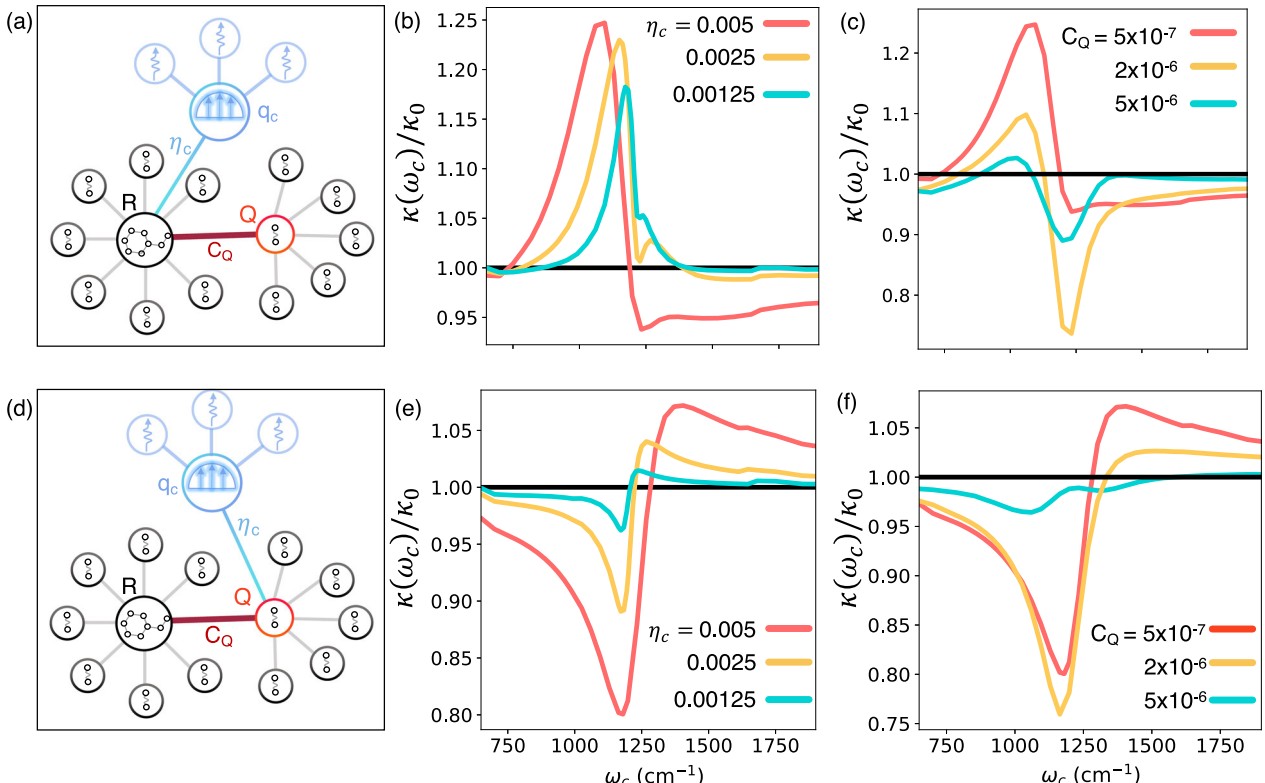

**Fig. 4 | Cavity modification of chemical reactivity in complex molecular systems. a** Schematic illustration of a molecular system embedded in a structured environment and coupled to a lossy cavity. Here, $R$, $q_c$, and $Q$ are coordinates of the reaction coordinate, cavity photon mode, and a spectator mode, respectively. Cavity modification of chemical reactivity at (**b**) various light–matter coupling values $\eta_c$ with fixed coupling between the reaction coordinate and spectator mode

of $C_Q = 5 \times 10^{-7}$ a.u. and at various values of (**c**) $C_Q$ with fixed $\eta_c = 0.005$ a.u. **d** Similar to (**a**) but with the cavity coupling to the spectator mode $Q$. **e**, **f** are similar to **b** and **c** but for the model system illustrated in (**d**). The cavity lifetime is set to $\tau_c = 1000$ fs. Here, $\kappa(\omega_c)$ is the cavity-modified rate constant as a function of the photon frequency $\omega_c$ and $\kappa_0$ is the rate constant outside the cavity. Source data are provided as a Source Data file.

$Q$, as schematically illustrated in Fig. 4a, d, respectively. Out-of-cavity absorption spectra are provided in Supplementary Note 2 for the six different sets of model parameters considered.

In Fig. 4b, we present the effect of the cavity coupling, at a fixed value of the coupling to the spectator mode $C_Q = 5 \times 10^{-7}$ a.u., molecule–solvent coupling $\eta_s = 0.1\omega_b$ and a cavity lifetime of $\tau_c = 1000$ fs, on the frequency-dependent chemical reaction rate for the cavity setup illustrated in Fig. 4a. Here, several competing effects play a role in modifying chemical reactivity. First, directly coupling $R$ to the cavity (which is coupled to a set of dissipative modes $\{Q_k\}$) increases the effective environmental friction, leading to a resonant enhancement of the reaction rate, which peaks around 1140 cm$^{-1}$ as discussed previously and shown in Figs. 2, 3. Secondly, coupling to the cavity splits the molecular vibrational levels (creating polariton states), pushing them further away from the sharp peak of the ($\omega_Q = 1200$ cm$^{-1}$) structured spectral density (red solid line in Fig. 1b), leading to a suppressed interaction between $R$ and $Q$. In other words, coupling to a cavity can drive the reaction coordinate away from resonance with the relevant solvent/spectator modes, thereby reducing the effective environmental friction.

Further, note that $\omega_0 \approx 1140$ cm$^{-1}$ is slightly lower than $\omega_Q = 1200$ cm$^{-1}$. As a result, when $\omega_c < \omega_0$, the chemical reaction rate is enhanced due to the enhanced interaction between $R$ and $Q$ as $|v_2, 0\rangle$ is pushed up energetically from below. On the other hand, for $\omega_c > \omega_0$ the molecular vibrational states (i.e., the lower polariton states) are pushed down, away from $\omega_Q$, reducing the interaction between $R$ and $Q$. In the Supplementary Information, we consider the opposite scenario where $\omega_Q < \omega_0$, such that the shape of $\kappa(\omega_c)$ is reversed, i.e., suppression at lower $\omega_c$ and enhancement at higher $\omega_c$, supporting this picture.

At higher light–matter coupling strengths, due to the larger Rabi-splitting, the suppression of interactions between $Q$ and $R$ becomes more important. Here, the photon frequency-dependent rate constant at relatively high light–matter coupling, $\eta_c = 0.005$ (red solid line in Fig. 4b), shows suppression at higher photon frequencies but enhancement at lower photon frequencies. However, at relatively weaker light–matter interactions, enhancement of the chemical rate, primarily originating from the effective increase in environmental friction, is observed. This can be seen for $\eta_c = 0.0025$ and $\eta_c = 0.00125$ a.u. depicted by the yellow and green lines in Fig. 4b, respectively. The chemical rate profile at $\eta_c = 0.00125$ (green solid line in Fig. 4b) is similar to that of the unstructured case (when $C_Q = 0.0$) shown in Fig. 3b with the same cavity lifetime $\tau_c = 1000$ fs, which is in harmony with the reasoning presented above.

Figure 4c illustrates the cavity modification of the chemical rate at various $C_Q$ and at constant $\eta_c = 0.005$ and $\eta_s = 0.1\omega_b$. The two competing effects and their relative importance depends on the value of $C_Q$. At higher $C_Q = 2 \times 10^{-6}$ a.u. (yellow solid line in Fig. 4c), the suppressed interaction between $Q$ and $R$ plays a more important role, which results in a sharp suppression of the reaction rate around 1200 cm$^{-1}$. Interestingly, at even higher $C_Q = 5 \times 10^{-6}$ a.u., the extent of cavity suppression of chemical reactivity becomes smaller. Note that with $C_Q = 5 \times 10^{-6}$ a.u., the system is just past the Kramers turnover point (see Fig. 1d). As a result, the decrease in the effective environmental friction (due to the light–matter Rabi-splitting) does not provide a significant modification of the chemical reaction rate.

Finally, in Fig. 4d–f, the cavity modification of chemical reactivity is investigated when coupling the cavity radiation mode to the

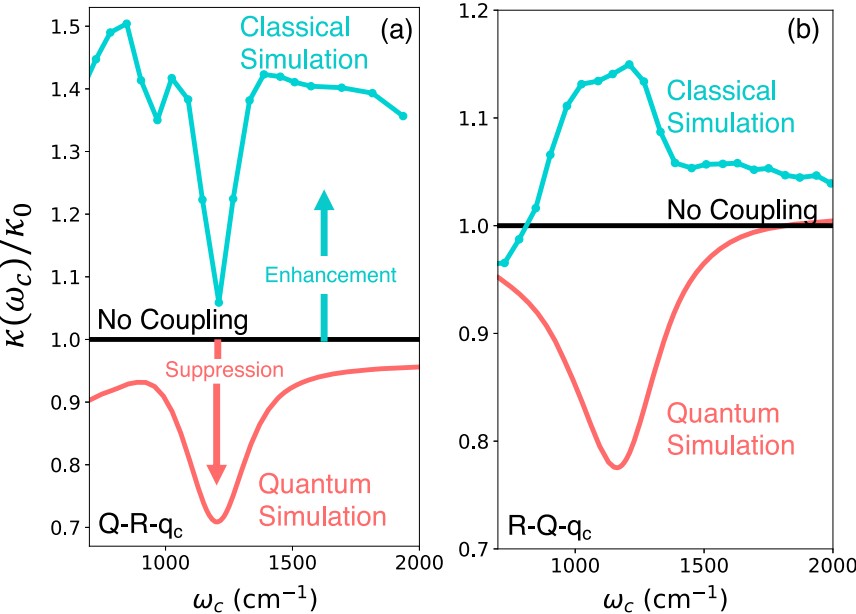

**Fig. 5 | Resonant cavity suppression of chemical kinetics under strong (post-turnover) solvent-molecule interactions.** The chemical rate constant $\kappa(\omega_c)$ as a function of the cavity photon frequency $\omega_c$ for a molecular system embedded in a structured environment (same as Fig. 4) when the cavity photon mode coordinate $q_c$ is coupled to **a** the reaction coordinate $R$ (illustrated in Fig. 4a) or **b** to the spectator mode $Q$ (illustrated in Fig. 4d) computed using exact quantum dynamics (red solid line) and classical dynamics (dashed cyan line) compared with the chemical rate constant in the absence of the cavity (black solid line). Here, $\kappa_0$ is the chemical rate constant outside cavity. The molecule-spectator mode coupling $C_Q = 5 \times 10^{-6}$ a.u. places the system in the post-turnover regime (using an unstructured solvent friction $\eta_s = 0.1\omega_b$). The cavity lifetime is set to $\tau_c = 1000$ fs and the light–matter coupling is set to $\eta_c = 0.01$ a.u. Source data are provided as a Source Data file.

spectator mode $Q$, namely, using $\hat{\mu} = \hat{Q}$ in $\hat{H}_{\mathrm{cav}}$. The cavity-molecule setup is schematically illustrated in Fig. 4d. Coupling the spectator mode to the cavity leads to the creation of polaritonic normal modes[19], which results in the splitting of the peak in the structured spectral density $J_S(\omega)$ in Fig. 1b. Due to this effect, the peak of the spectral density can be shifted away from the $|v_L\rangle \rightarrow |v_2\rangle$ transition, leading to a suppressed interaction between the molecule and its environment (which now also include the cavity mode). Note that while in the present work, we do not directly address the issue of collectivity, the model presented here is structurally the same as an idealized model for studying collective effects[16,40]. That is, the mode $Q$ can be viewed as a collective solvent coordinate, such that the light–matter coupling $\eta_c$ scales with $\sqrt{N}$ where $N$ is the number of solvent modes directly coupling to the molecule.

In Fig. 4e, the effect of increasing the light–matter coupling $\eta_c$ between $Q$ and $q_c$ is investigated. Overall, increasing the light–matter coupling results in a larger splitting of the spectral density, leading to a larger amount of suppression, with a dip occurring around $\omega_Q = 1200$ cm$^{-1}$. As mentioned previously, since $\omega_Q$ is slightly higher than the $\omega_0 = E_2 - \bar{E}_0 \approx 1140$ cm$^{-1}$, the peak of the spectral density $\omega_Q$ is pushed closer to the molecular energy gap $\omega_0$ for $\omega_c > \omega_Q$. This results in increased interaction between the molecule and its environment at higher photon frequencies (>1250 cm$^{-1}$), leading to an enhancement of chemical reactivity in Fig. 4e in the pre-turnover, weak-coupling regime.

In Fig. 4f we explore the effect of increasing $C_Q$ on cavity-modified chemical reactivity. Similar to Fig. 4c, we observe that an initial increase in $C_Q$ (to $2 \times 10^{-6}$ a.u., yellow solid line) leads to further modification (i.e., suppression) of chemical kinetics. This behavior can be understood by looking at how the reaction rate is modified as a function of $C_Q$, as presented in Fig. 1d. For very low $C_Q \rightarrow 0$, the slope of the rate $\kappa$ is smaller than that of the slope at relatively higher values $C_Q = 2 \times 10^{-6}$ a.u. This implies that changes in environmental friction for $C_Q = 5 \times 10^{-7}$ a.u. lead to small changes in the chemical reaction rate

than they do for $C_Q = 2 \times 10^{-6}$ a.u. This effect explains why chemical rate modifications are relatively milder for $C_Q = 5 \times 10^{-6}$ a.u., which is close to the turnover region (see Fig. 1d). Note that for $C_Q = 5 \times 10^{-6}$, we also see a splitting of the rate profile with peaks corresponding to the normal modes formed by strongly coupling the $R$ and $Q$ modes (a Rabi-splitting due to coupling between two nearly resonant vibrational modes).

Overall, the results presented in Fig. 4 demonstrate that it is possible to observe both resonant enhancement or resonant suppression depending on how the cavity couples to the matter subsystem. The cavity can either enhance (such as in 4a–c) or suppress (such as in 4d–f) the effective environmental friction felt by the molecule. These effects originate from quantum light–matter hybridization, and consequently, the chemical rate as a function of photon frequency exhibits sharp resonances. Our results point to the possibility of observing both suppression and enhancement in the same frequency scale, depending on photon frequency. We note that such behavior (observing both suppression and enhancement) has not yet been observed in present experiments[1–3,6,8,10].

Finally, in Fig. 5, we demonstrate that sharp resonant suppression of the ground state chemical kinetics can be observed even for strong molecule–solvent interactions (post-turnover). This is important since many systems are experimentally expected to be found in this strong-coupling (post-turnover) regime. Here, the cavity mode $q_c$ is coupled to $R$ in Fig. 5a or to $Q$ in Fig. 5b as schematically illustrated in Fig. 4a, d, respectively. Importantly, here we set $C_Q = 5 \times 10^{-6}$ a.u. (and $\eta_s = 0.01\omega_b$) that places the environmental friction in the post-turnover regime. Despite the strong coupling (post-turnover), significant suppression of chemical reactivity, in the range of 25–30%, is achieved in both scenarios by using a slightly higher light–matter coupling $\eta_c = 0.01$ a.u. (corresponding to a quantization volume of $V \sim 10$ nm$^3$). In both cases, our exact quantum simulations show a sharp resonant suppression, just as

observed in experiments[1,2,8] and in contrast to all previous theoretical studies that show a weak and broad cavity suppression profile[14–16,20,22,27,29–31]. Here we compare the rate constant computed from the quantum simulations (red solid line) with Grote–Hynes classical rate theory[14,23,24], which predicts minute and broad suppression of chemical reactivity. Note that the quantum results presented in Fig. 5a show a cavity frequency-independent suppression in addition to the resonant cavity frequency-dependent suppression. This cavity frequency-independent suppression is due to the renormalization (rescaling of $\Delta \to \Delta \cdot \langle \exp[\pm i\eta_c \sqrt{2/\omega_c}\hat{p}_c\Delta\mu]\rangle$ due to the difference in permanent dipoles) of the tunneling parameter when coupling $q_c$ directly to the reaction coordinate $R$, as explained in Fig. 3.

In this work, we present an in-depth study of cavity modification of chemical reactivity in a model that contains most of the features needed to realistically describe how light–matter coupling may alter chemical reaction rates. Crucially, our work treats all degrees of freedom quantum mechanically, and does so in a numerically exact manner. This enables a detailed understanding of the factors that enhance or reduce reaction rates and the explicit role played by Rabi-splitting and resonances on these effects. Our work highlights the fact that an accurate description of this modification requires an accurate treatment of the energy transfer processes arising in the presence of strong light–matter interactions. In the strong coupling regime, classical[70,71] and mixed quantum-classical methods[72,73] are known to have difficulties describing such processes quantitatively. That is, the quantitative (and in some cases even qualitative) description of the modification of the rate of a chemical reaction inside an optical cavity requires a full treatment of the quantum nature of the cavity radiation mode, solvent modes, and molecular vibrations.

The lessons from the study of vibrational energy relaxation, which is a crucial step in the polaritonic modification of chemical reaction rates, are critical for understanding when we can expect a breakdown of the classical treatment for reaction rates inside a cavity. We expect that there are two potentially simultaneously operative regimes where classical calculations will be in error:

- When a Golden Rule-level of theory for rates cannot be applied. As shown by ref. 72 the Golden Rule vibrational relaxation rate for a purely harmonic, purely classical system is identical to that of a purely harmonic quantum system. However, this agreement will breakdown sharply when one needs to go beyond the Golden Rule to capture the relaxation rates. We expect that this is the case in our system as the rate is governed by resonances, obviating low-order perturbation theory.

- As shown by ref. 70 when a system is nonlinear, then purely classical theories of vibrational relaxation, while potentially preferable to mixed quantum-classical ones, are not quantitatively accurate even in the Golden Rule limit, especially for high-frequency system modes, regardless of the bath frequencies. In the case studied by Egorov and Berne, the nonlinearity comes from the form of the system-bath coupling, while in our case, it comes from the form of the potential energy curve along the reaction coordinate. It is simple to show that these cases will behave similarly by writing our Hamiltonian on the basis of the vibrational eigenfunction of the system. On this basis, the system-bath term will couple all eigenstates, and not just ones that differ by one vibrational quantum, just as in the model of ref. 70. Note that previous studies[29–31] that also used a quantum mechanical treatment of cavity and molecule but ignored the solvent failed to capture the features of the cavity modification of chemical kinetics that we expose here. Without a proper account of the dissipation produced by the solvents as well as quantum treatment of cavity loss, a complete picture of how a cavity modifies chemical kinetics cannot be obtained.

**Table 1 | Summary of the observed cavity modification of the chemical reactivity for the model systems investigated in this work**

| Cavity properties | Solvent properties | Effect observed |
|---|---|---|
| Lossy $q_c - R$ | Unstructured Tunneling dominated | Broad suppression |
| Lossy $q_c - R$ | Unstructured Energy diffusion-limited | Sharp enhancement |
| Perfect $q_c - R$ | Unstructured Energy diffusion-limited | No effect |
| Lossy $q_c - R$ | Structured Energy diffusion-limited | Sharp enhancement and/or suppression |
| Lossy $q_c - Q$ | Structured Energy diffusion-limited | Sharp enhancement and/or suppression |
| Lossy $q_c - R$ | Structured Spatial diffusion-limited | Sharp suppression |
| Lossy $q_c - Q$ | Structured Spatial diffusion-limited | Sharp suppression |

The first column describes the properties of the cavity, such as if it is lossy (has a finite cavity lifetime) or perfect (cavity lifetime is infinite) and to which of the matter degrees of freedom it is coupled to. Here, $q_c - R$ and $q_c - Q$ represents a cavity photon mode coupled to the reaction coordinate $R$ or to the spectator mode coordinate $Q$, respectively. The second column describes the solvent properties, such as the structure of its spectral density (structured or unstructured, see Fig. 1b) and if the solvent coupling places the system in the tunneling-dominated, energy or spatial diffusion-limited regime. The third column describes how coupling to cavity modifies chemical reactivity, such as enhancement and/suppression, and if the photon frequency dependence of the cavity-modified chemical rate is sharp or broad.

In Table 1, we summarize how coupling to a cavity influences the chemical reactivity of some of the models studied in this work. Here $q_c - R$ and $q_c - Q$ signifies coupling the cavity to the reaction coordinate or to the spectator mode, respectively. Overall, for unstructured spectral densities, we observed sharp resonant cavity modification of the chemical rate in the energy diffusion-limited for finite cavity lifetime. For structured spectral densities, we find that it is possible to obtain sharp suppression or enhancement of the rate, depending on whether the cavity coupling increases or decreases the solvent-molecule interactions. When some solvent vibration modes have frequencies close to molecular vibrational transitions, the net coupling between the molecule and the solvent is modified by either coupling the cavity to the molecular vibrations or to the solvent mode. Coupling the cavity radiation mode to molecular vibration splits the molecular vibrational states, thereby either moving them closer or further away from the solvent modes. A similar situation occurs when coupling the cavity radiation mode to the solvent modes. Whether this leads to enhancement or suppression (or both) depends on whether solvent-molecule coupling strength places the system in the energy or spatial diffusion-limited regimes, and on how molecule–solvent interactions are modified. Note that because such effects originate from the requirement of state splitting, something that only takes place near resonance, we observe sharp photon frequency dependence of the cavity-modified rate.

Remarkably, we find that the features found in our simulations are similar to those seen in experiments carried out at finite molecular concentrations. We show that alteration of reactivity is subtle and depends sensitively on factors such as which modes couple to the cavity and the rate of cavity loss. Many aspects of cavity-modified reactivity that we expose in this work have not been previously captured by theoretical studies. Previous work that treats the molecular and radiation degrees of freedom classically have found that cavity modification of chemical kinetics is photon frequency dependent and maximum suppression occurs when the photon frequency is close to the barrier frequency[14–17] in the spatial diffusion-limited regime (exhibiting suppression) or when photon frequency is close to well frequency in the energy diffusion-limited

regime[17,20,27]. However, in these works, the rate profile is much broader (~500–5000 cm⁻¹) than the corresponding absorption profiles of the bare molecular system and the peak modification typically appeared comparatively far from the relevant vibrational transition frequencies. In contrast, we find that most enhancement or suppression of chemical reactions occurs near resonance with molecular vibrational frequencies, and that these features are sharp (e.g., on the order of a few hundred wavenumbers in width at most). Past classical and semiclassical calculations could produce only a very weak and very broad suppression of the rate in the regime past the Kramers turnover, we find that sizable and sharp suppression may occur in this regime once an exact, fully quantum approach is used.

One obvious important aspect missing from our model is the collective effects induced by the cavity and by direct coupling between molecules. It is natural to assume that collective effects will increase the magnitude of the features presented here without qualitatively altering the behavior. However, explicit many-molecule quantum mechanical calculations are needed to confirm this expectation. An exact quantum mechanical treatment of such a situation is likely prohibitively expensive for realistic models. On the other hand, the results presented here may be used as benchmarks for approximate quantum mechanical approaches aimed at treating such collective effects. We plan to carry out such studies in future work.

## Methods

### Hierarchical equations of motion

All Quantum mechanical simulations were performed using the hierarchical equations of motion (HEOM) approach. This well-established open-quantum system dynamics method provides an exact description of the dynamics of a quantum system that is linearly coupled to a set of $N$ harmonic baths[74–76]. The Hamiltonian for such a system may be written as

$$\hat{H} = \hat{H}_S + \sum_{i=1}^{N} \hat{H}_{B,i} + \sum_{i=1}^{N} \hat{S}_i \hat{B}_i, \tag{6}$$

where $\hat{H}_S$ and $\hat{H}_{i,B} = \sum_{\alpha} \omega_{i,\alpha} \hat{a}_{i,\alpha}^\dagger \hat{a}_{i,\alpha}$ are the bare system and $i$th harmonic bath Hamiltonians, respectively, $\hat{S}_i$ is a system operator, and $\hat{B}_i = \sum_{\alpha} g_{i,\alpha}(\hat{a}_{i,\alpha}^\dagger + \hat{a}_{i,\alpha})$ is the $i$th bath coupling operator. The frequencies, $\omega_{i,\alpha}$, and coupling constants, $g_{i,\alpha} = c_{i\alpha}/\sqrt{2\omega_{i\alpha}}$ for bath $i$ are fully specified by the bath spectral density $J_i(\omega) = \pi \sum_{\alpha} g_{i,\alpha}^2 \delta(\omega - \omega_{i,\alpha})$, which uniquely captures the influence of the bath on the system at a given temperature.

The HEOM approach makes use of the Feynman-Vernon influence functional to obtain the exact quantum dynamics of the system in terms of an infinite set of auxiliary density operators (ADOs), $\hat{\rho}_{\boldsymbol{n}}(t)$[74,76]. These ADOs are indexed by vectors of integers of length $K_i$ for each bath, that is $\boldsymbol{n} = (n_{0,0}, n_{0,1}, \ldots, n_{0,K_0}, n_{1,0}, \ldots, n_{N,K_N})$. This infinite set of ADOs contains the system-reduced density operator as the element $\hat{\rho}_{\boldsymbol{0}}(t) = \mathrm{Tr}_B[\hat{\rho}(t)]$, where $\mathrm{Tr}_B$ denotes the partial trace over all bath degrees of freedom and $\hat{\rho}(t)$ is the density operator for the full system and bath. All other ADOs account for correlations between the system and bath degrees of freedom that arise from the system-bath coupling terms in Hamiltonian Eq. (6).

The ADOs evolve according to the infinite set of coupled ordinary differential equations[77]

$$\begin{aligned}
\frac{\mathrm{d}}{\mathrm{d}t} \hat{\rho}_{\boldsymbol{n}}(t) = &-i[\hat{H}_S, \hat{\rho}_{\boldsymbol{n}}(t)] - \sum_{i=1}^{N} \sum_{k=1}^{K_j} n_{i,k} \nu_{i,k} \hat{\rho}_{\boldsymbol{n}}(t) \\
&- i \sum_{i=1}^{N} \sum_{k=1}^{K_j} \left( \sqrt{n_{i,k}+1} \mathcal{B}_{i,k}^- \hat{\rho}_{\boldsymbol{n}_{i,k}^+}(t) + \sqrt{n_{i,k}} \mathcal{B}_{i,k}^+ \hat{\rho}_{\boldsymbol{n}_{i,k}^-}(t) \right),
\end{aligned} \tag{7}$$

where $\boldsymbol{n}_{i,k}^\pm = (n_{0,0}, \ldots, n_{i,k} \pm 1, \ldots, n_{N,K_N})$, and here we have used a rescaled form for the ADOs. In Eq. (7) the Liouville space system operators, $\mathcal{B}_{i,k}^\pm$, are defined by

$$\mathcal{B}_{i,k}^- \hat{O} = \sqrt{|c_{i,k}|}[\hat{B}_i, \hat{O}] \tag{8}$$

$$\mathcal{B}_{i,k}^+ \hat{O} = \left( \frac{c_{i,k}}{\sqrt{|c_{i,k}|}} \hat{B}_i \hat{O} - \frac{\bar{c}_{i,k}^*}{\sqrt{|c_{i,k}|}} \hat{O} \hat{B}_i \right) \tag{9}$$

where $\nu_{i,k}$, $c_{i,k}$ and $\bar{c}_{i,k}$ are obtained from a decomposition of the bath correlation function,

$$C_i(t) = \frac{1}{Z_b} \mathrm{Tr}\left[ \hat{B}_i(t) \hat{B}_i e^{-\beta \hat{H}_{i,B}} \right], \tag{10}$$

and its complex conjugate into a sum of exponentials[76],

$$C_i(t) = \sum_{k=1}^{K_i} c_{i,k} e^{-\nu_{i,k} t} \tag{11}$$

$$C_i^*(t) = \sum_{k=1}^{K_i} \bar{c}_{i,k}^* e^{-\nu_{i,k} t}. \tag{12}$$

For an exact representation of the bath correlation function, it will typically be necessary to include an infinite number of terms in this expansion. Here we truncate to a finite number of terms $K_i$ for each bath by using a ($[K_i-1]/[K_i]$) Páde approximant of the Bose function[78,79].

In order to obtain a practical simulation method, it is typical to truncate this infinite series of ADOs. For the models considered in the main text, and in particular, those approaching Kramer's turnover regime, we were unable to converge the dynamics with respect to the size of the hierarchy using the commonly used strategies of truncating the hierarchy of ADOs at a fixed total excitation number[75], $L$, (including all ADOs satisfying the conditions $\sum_{i=1}^{N} \sum_{k=1}^{K_i} n_{i,k} \leq L$) or at a fixed total decay rate[80], $\nu_M$, (including all ADOs satisfying $\sum_{i=1}^{N} \sum_{k=1}^{K_i} \nu_{i,k} n_{i,k} \leq \nu_M$). We found that converged results could be obtained using a modified truncation strategy in which we include all ADOs satisfying the condition

$$\sum_{i=1}^{N} \sum_{k=1}^{K_i} \max(\nu_{i,k} n_{i,k}, \Gamma \nu_{\min}) \leq L \nu_{\min}, \tag{13}$$

where $\nu_{\min} = \mathrm{real}(|\nu_{i,k}|)$. This approach combines aspects of the two more commonly used approaches—it includes all ADOs:
- with a decay rate less than $\nu_M = L \nu_{\min}$
- and with up to $L/\Gamma$ excitations regardless of their decay rate

### Evaluation of the forward reaction rate

In order to evaluate the forward reaction rate, we have assumed that the system is initially in the reactant region with an initial density operator $\hat{\rho}(0) = \hat{\rho}_R$. The time-dependent reactant and product populations may be written as

$$P_R(t) = \mathrm{Tr}\left[ (1 - \hat{h}) \hat{\rho}(t) \right] \tag{14}$$

$$P_P(t) = 1 - P_R(t), \tag{15}$$

where $\hat{h}$ is the side operator that projects onto the reactant states, and is only a function of the reaction coordinate position operator, $\hat{R}$, in our work. Provided first-order kinetics provides a valid description of

the reaction process, then in the long-time limit, the reactant and product populations will evolve according to the kinetic equations[60,81,82]

$$\dot{P}_R(t) = -\kappa P_R(t) + \kappa' P_P(t) \tag{16}$$

$$\dot{P}_P(t) = \kappa P_R(t) - \kappa' P_P(t), \tag{17}$$

where $\kappa$ and $\kappa'$ are the forward and backward rate constants, respectively (and are related by $\kappa\langle P_R\rangle = \kappa'\langle P_P\rangle$, where $\langle P_R\rangle$ and $\langle P_P\rangle$ are the equilibrium reactant and product populations, and can be obtained from the steady-state solution of the HEOM[83]). Rearranging the expression for the forward rate constant, we have[37,60,81]

$$\kappa = \lim_{t\to\infty} \frac{\dot{P}_P(t)}{1 - P_P(t)/\langle P_P\rangle}, \tag{18}$$

where the limit $t\to\infty$ indicates that the kinetic description of the reaction process is only valid after some initial transient process.

We have considered two choices for the reactant density operator. The first, an uncorrelated (between system and bath), thermal density operator[60]

$$\hat{\rho}_R = \frac{1}{Z_R} e^{-\beta\hat{H}_S/2}(1-\hat{h})e^{-\beta\hat{H}_S/2} \otimes \frac{e^{-\beta\hat{H}_B}}{\text{Tr}\left[e^{-\beta\hat{H}_B}\right]}, \tag{19}$$

where $Z_R = \text{Tr}\left[e^{-\beta\hat{H}_S/2}(1-\hat{h})e^{-\beta\hat{H}_S/2}\right]$, allows for the direct application of the HEOM approach discussed above. The second, a correlated, thermal reactant density operator[37,56]

$$\hat{\rho}_R = \frac{\hat{\rho}_{SS}(1-\hat{h})}{\text{Tr}\left[\hat{\rho}_{SS}(1-\hat{h})\right]}, \tag{20}$$

where we have used the steady-state solution of the HEOM, $\hat{\rho}_{SS}$, as the thermal equilibrium state[83], requires the evaluation of the HEOM steady state before evaluation of the rate constant. The short-time transient dynamics depend on the choice of the initial reactant density operator; however, we have found that the long-time plateau value of Eq. (18) is independent of the choice of initial density operator for the models considered in this work.

In principle, the HEOM steady state can be obtained by propagating an uncorrelated initial approximation to the thermal density operator

$$\hat{\rho}_0 = \frac{e^{-\beta(\hat{H}_S + \hat{H}_B)}}{\text{Tr}\left[e^{-\beta(\hat{H}_S + \hat{H}_B)}\right]} \tag{21}$$

in time according to Eq. (7), until a steady state is reached. While this strategy can prove successful in the case of weakly coupled baths, in general, such an approach may require excessively long simulation times. Alternatively, the steady state can be obtained by solving the linear system of equations defined by Eq. (7), subject to the steady-state condition,

$$\frac{d}{dt}\hat{\rho}_n(t) = 0. \tag{22}$$

Here, we have solved this resultant large, sparse linear system of equations using the BiCGStab(l) approach[84].

IR spectra were computed from the dipole-dipole correlation function according to the equation[85]

$$\alpha(\omega)n(\omega) \propto \lim_{t_f\to\infty} \omega(1 - \exp(-\beta\omega)) \int_{-\infty}^{\infty} C_{\mu\mu}(t)e^{-i\omega t}. \tag{23}$$

Here,

$$C_{\mu\mu}(t) = \frac{\text{Tr}\left[\hat{\mu}(t)\hat{\mu}\hat{\rho}_{SS}\right]}{\text{Tr}\left[\hat{\rho}_{SS}\right]} \tag{24}$$

is the dipole-dipole correlation function obtained using the steady-state solution to the HEOM as the initial thermal density operator. This correlation function was obtained following the procedure outlined in ref. 86.

## Classical dynamics

We rewrite the total Hamiltonian provided in Eq. (1) as

$$\begin{aligned}\hat{H} &= \hat{H}_{mol} + \hat{H}_{solv} + \hat{H}_{cav} + \hat{H}_{loss} \\ &= \hat{H}_{mol} + \hat{H}_{cav} + \hat{H}_Q + (\hat{H}_{loss} + \hat{H}'_{solv}) \\ &= \hat{H}_0 + (\hat{H}_{loss} + \hat{H}'_{solv}) \end{aligned} \tag{25}$$

where $\hat{H}_Q = \frac{\hat{P}_Q^2}{2} + \frac{1}{2}\omega_Q^2\left(\hat{Q} + \frac{C_Q\hat{R}}{\omega_Q^2}\right)^2$, $\hat{H}_0 = \hat{H}_{mol} + \hat{H}_{cav} + \hat{H}_Q$ and $\hat{H}'_{solv} = \hat{H}_{solv} - \hat{H}_Q$. Using the this Hamiltonian we write the classical equation (generalized Langevin equation) of motion for $R$, $Q$, and $q_c$ as

$$\ddot{R}(t) = -\frac{\partial H_0}{\partial R} + \int_0^t \dot{R}(\tau)\cdot\eta_B(t-\tau) + \xi_B(t), \tag{26}$$

$$\ddot{Q}(t) = -\frac{\partial H_0}{\partial Q} + \int_0^t \dot{Q}(\tau)\cdot\eta_Q(t-\tau) + \xi_Q(t), \tag{27}$$

$$\ddot{q}_c(t) = -\frac{\partial H_0}{\partial q_c} + \int_0^t \dot{q}_c(\tau)\cdot\eta_c(t-\tau) + \xi_c(t). \tag{28}$$

The corresponding friction kernel is given as

$$\begin{aligned}\eta_B(t) &= \sum_i \frac{C_i^2}{\Omega_i^2}\cos(\Omega_i t) = \frac{2}{\pi}\int_0^\infty d\Omega \frac{J_B(\Omega)}{\Omega}\cos(\Omega_i t), \\ &= \frac{2}{\pi}\Gamma\Lambda_s \int_{-\infty}^\infty d\Omega \frac{\cos(\Omega t)}{\Omega^2 + \Gamma^2} = 2\Lambda_s e^{-\Gamma t}.\end{aligned} \tag{29}$$

Similarly, we have $\eta_Q(t) = 2\lambda_s e^{-\gamma t}$ and $\eta_c(t) = 2\lambda_L e^{-\gamma_L t}$. We rewrite the equation of motion in the extended phase space (adding new degrees of freedom $S_B$, $S_Q$, and $S_c$) as[87]

$$\dot{P} = -\frac{\partial H_0}{\partial R} + S_B(t), \tag{30}$$

$$\dot{P}_Q = -\frac{\partial H_0}{\partial Q} + S_Q(t), \tag{31}$$

$$\dot{p}_c = -\frac{\partial H_0}{\partial q_c} + S_c(t), \tag{32}$$

$$\dot{S}_B = -\Gamma S_B(t) - 2\Lambda_s P(t) + 2\Gamma\sqrt{\Lambda_s/\Gamma\beta}\dot{W}_B(t), \tag{33}$$

$$\dot{S}_Q = -\gamma S_Q(t) - 2\lambda_s P_Q(t) + 2\gamma\sqrt{\lambda_s/\gamma\beta}\dot{W}_Q(t), \tag{34}$$

$$\dot{S}_c = -\gamma_L S_c(t) - 2\lambda_L p_c(t) + 2\gamma_L \sqrt{\lambda_L/\gamma_L \beta} \dot{W}_c(t). \tag{35}$$

Here each $\{W_i\}$ is a standard Wiener process. The equations of motion are then evolved using a velocity Verlet-type algorithm[87]. In order to compute the chemical rate, we compute the transmission coefficient from the flux-side correlation function formalism[88], similar to recent works[14,15,20,67], as follows

$$\kappa(t) = \frac{\langle \mathcal{F}(0) \cdot h[R(t) - R_\ddagger]\rangle}{\langle \mathcal{F}(0) \cdot h[\dot{R}_\ddagger(0)]\rangle}, \tag{36}$$

where $h[R - R_\ddagger]$ is a Heaviside function. Here, the dividing surface $R_\ddagger = 0$ separates the reactant and the product and $\langle \ldots \rangle$ represents the canonical ensemble average (with a constraint on the dividing surface, which is enforced by $\delta[R(t) - R_\ddagger]$ inside $\mathcal{F}(t)$). Further, $\dot{R}_\ddagger(0)$ represents the initial velocity of the reaction coordinate on the dividing surface that is obtained by sampling a classical Maxwell–Boltzmann distribution at $T = 300$ K. The reaction coordinate is initialized at the dividing surface $R(t = 0) = 0$ while the extended variables $S_c$ and $S_B$ are initialized from a gaussian distribution with the distribution widths $\sqrt{2\Lambda_s/\beta\Delta t}$ and $\sqrt{2\lambda_s/\beta\Delta t}$ (time step $\Delta t = 10$ a.u.), respectively[87]. A total of 1,500,000 configurations are released from the dividing surface, with the initial momentum $p_c$ and $P$ randomly sampled from the classical Maxwell–Boltzmann distribution. Each trajectory is propagated for 8 ps such that the flux-side correlation function has plateaued.

## Reporting summary
Further information on research design is available in the Nature Portfolio Reporting Summary linked to this article.

## Data availability
The source data underlying all figures are available at ref. 91. Source data are provided with this paper.

## Code availability
The source code that support the findings of this study are available at refs. 92,93.

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

## Acknowledgements

This work was supported by NSF-1954791 (A.M. and D.R.R.) and by the Chemical Sciences, Geosciences, and Biosciences Division of the Office of Basic Energy Sciences, Office of Science, US Department of Energy (L.P.L. and D.R.R.). This work used the Extreme Science and Engineering Discovery Environment (XSEDE), which is supported by National Science Foundation grant number ACI-1548562 (allocations: TG-CHE210085). Specifically, it used the services provided by the OSG Consortium[89,90], which is supported by the National Science Foundation awards #2030508 and #1836650.

## Author contributions

L.P.L., A.M., and D.R.R. designed the research. L.P.L. performed the exact quantum dynamics simulations. A.M. performed classical simulation. L.P.L. and A.M. set up the model for the molecule-cavity system. L.P.L., A.M., and D.R.R. wrote the manuscript.

## Competing interests

The authors declare no competing interests.
