## [Peer Review File · Nature Communications]

Quantum dynamical effects of vibrational strong coupling in chemical reactivityREVIEWER COMMENTS

Reviewer #1 (Remarks to the Author):

Summary

This work reports on quantum dynamics calculations on a model of a reactive molecule coupled to a cavity mode. The molecule consists of a double-well potential linearly coupled to a dissipative bath, and eventually also linearly coupled to a spectator mode with its own sub-bath. The cavity consists of a single cavity mode linearly coupled (via PF-Hamiltonian) to either the reactive coordinate or to the spectator mode and also features a linear bath modeling cavity losses.

The goal of the study is to provide fully quantum mechanical benchmark results for the modification of thermal chemical reactions in the strong vibrational coupling regime. The rationale for the fully quantum, model-based approach is that up to now studies have been either based on analytical theories (Grote-Hynes and similar) applied to extremely simple Hamiltonians, or based on classical dynamics on realistic potential surfaces.

Key results

1. The modification of the chemical rate occurs sharply at resonances of the molecular system with the cavity mode. Enhancement of chemical rates is related to the stabilization of the products caused by energy transfer to the cavity.
2. In the strong molecule-solvent interaction regime, the cavity can still modify the rate, but in this case, reducing it.
3. It makes a difference whether the cavity couples directly to the reactive coordinate or to the spectator mode. Both enhancement and suppression can take place depending on the specific cavity frequency.

Significance

The paper is in an area of active research, where experimental results on the modification of rates are not yet fully understood, in part because these experiments are performed in complex systems where accurate computational modeling is extremely challenging or simply impossible. Therefore, theoretical contributions that provide new evidence and consolidate existing ideas are important for the advancement of the field.

In this respect, Point 1 above is the conclusion also reached in Ref. 16 on the basis of trajectory-based classical simulations of the chemical rate in an ab initio potential surface. There, converged reactive-flux calculations find a sharp resonance when the cavity is resonant with a coordinate strongly coupled to the reaction coordinate, similar to the Q_2 coordinate here. Also Point 2 is in part seen in Ref. 16, although there, the modulation with the cavity frequency was not explored.

The significance of the work thence rests very much on the fully quantum nature of the calculations compared to previous studies, although at the cost of introducing a simple model based on one reactive coordinate and linearly coupled oscillators. To better assess this significance, I would like to ask the authors to comment on the following points:

1. I have some problems with statements like in line 114: "These effects emerge from fundamental quantum light-matter interactions and cannot be fully captured with simple classical or semiclassical descriptions of light and matter."

Is this really so? Of course, the fundamental and ultimately correct theory to describe matter, light, and light-matter is quantum mechanics, but I do not agree with the implied message: because we are dealing with a hybrid light-matter system, then only quantum mechanics works and can be trusted. If we look into the Hamiltonian, the light-matter coupling terms are of simpler nature and complexity (i.e. lower order of couplings) than the intramolecular potential surfaces of most polyatomic molecular systems. What determines to a large extent how much a CM treatment deviates from QM is the density of quantum states at the total energy of the system. Classical rate calculations including recrossing are extremely successful at capturing chemical rates at room temperature for reactions in the gas phase and in solution. Missing quantum effects are often due to comparatively small tunneling corrections and zero-point effects. I do not see the cavity making a fundamental difference in this respect. It is a further oscillator at a typical frequency for a vibrational mode.

Hence, the reported differences between the CM and QM description of the rate, and the different widths of the rate resonance, can probably be attributed to the large weight of the tunneling mechanism in this model, with a low and narrow barrier. The tunneling mechanism is more sensitive to resonances, and hence to alterations made by the cavity, than over-the-barrier thermally-activated pathways.

It would be useful to consider this possibility and, if feasible, try to simulate a less tunneling-dominated scenario. The asymmetric potential in Figure S3, but starting from the deep well instead, might suppress deep tunneling and be an interesting test case.

2. How realistic is a model based on linear couplings only? Although exact HEOM calculations are possible on it, it may oversimplify other aspects of the dynamics. For example, its dynamics may be much less chaotic than those of realistic potentials with higher-order anharmonic couplings. This might explain the lack of an effect in the absence of cavity losses (Fig. 3b, text in lines 345 and below).

3. In lines 571-574 one reads the statement that the enhancement or suppression of chemical rates near resonance with molecular vibrations had not been captured by previous theoretical studies. This finding has been reported on the basis of classical flux calculations before. In the present contribution, the authors report this resonant behavior also in the strong solvent friction regime, which had been seen as well, but not studied as a function of the cavity frequency. Maybe these lines could be rewritten to more accurately reflect the current state of understanding.

4. Is the formation of the Rabi splitting in the absorption spectrum really determining the cavity effect as shown in Fig. 2d and discussed in lines 262 and below? On the one hand, the formation of polaritonic resonances can be seen in fully classical treatments [e.g. Tao et. al, *Angew. Chem.* 60, 15533 (2021)]. On the other hand, this viewpoint is not adopted when considering the coupling of R with Q , although exactly the same consideration could have been made.

Minor points --

1. What is the temperature in the quantum dynamical HEOM simulations? I could not identify this parameter in the main article or the methods section. The supporting information reports 300 K, but only for the classical case.

2. Line 202: Shouldn't Fig 1b be Fig 1c?

Summary --

In my opinion, this is an interesting contribution with insightful elements. Its technical quality and validity are without doubt of a high standard. I am not yet fully convinced that its significance matches the Nat. Comm. criteria. Its main conclusions solidify and partially extend previous theoretical and computational observations. It builds strongly upon the notion that quantum effects are fundamental in the vibrational strong coupling. The way this is argued does not fully convince me, although this can maybe be remedied.

Reviewer #2 (Remarks to the Author):

The authors present a theoretical model for chemical reactions under vibrational strong coupling (VSC). This means a molecular vibration has been entangled to the confined electromagnetic field of a photonic cavity, resulting in the creation of vibrational light-matter states, a.k.a. (vibro-)polaritons. This is an exciting new field on the border of physics and chemistry as it promises to develop an entirely new approach for controlling chemical reactions. The big challenge the field is now facing, is to understand the mechanism behind the observed modifications of reactions executed in a resonant cavity.

I am active in the field as an experimentalist for many years now, in which I consider myself an expert. Therefore, I will mainly focus on the value of the proposed model to explain VCS chemistry experiments. The implementation of the computational approach needs to be verified by another expert.

To start with the positive, the authors bring forward an interesting attempt for a fully quantum-mechanical model for chemistry under VSC and have succeeded in calculating a wealth of characteristics. Without a doubt, this is testament to their strength as a theoretical group. The focus they put on including dissipative interactions with spectator vibrations and the solvent bath is interesting and constitutes an important, original contribution to the field. Several trends they observe, are counter-intuitive but fascinating. Their overall explanation for VSC effects based on considering VSC modification of the interaction of the molecule with its environment seems like a strong candidate to answer the outstanding question in the field. The manuscript is also quite clearly written (although there are still a few odd sentences to be found, likes lines 275-278 and line 518-521).

However, I have concerns from 2 different perspectives which make me doubt the value of the proposed model for polaritonic chemistry.

1) Modeling of a reaction

In order to use an approach which allows the authors to make their quantum mechanical model exactly solvable some serious simplifications were used in constructing their Hamiltonian, esp. regarding modeling of the reacting molecule. These choices are not motivated in the text, which is a serious weakness. Citing previous work which uses such simplifications and obtain useful results would heavily strengthen the case the authors want to make.

The most serious simplification is only considering a single reacting molecule. In strong coupling this will not give you the polaritonic dark states. These are generally considered to have a major influence as the Jaynes-Cumming-Tavis model predicts that there should be always so many of them: $N-1$ dark states when N molecules are coupled. In vibrational strong coupling, you couple typical 10^5 - 10^6 dipoles/molecules into a set of coherent polaritons states. Many theoreticians in the field of organic polaritonics now consider the many molecules case needs to be considered in order to successfully explain and predict effects on chemical reaction and other material properties. This approximation might also explain why only relatively small relative effects are found here (<

30 %), while much larger effects have been measured (4-to-5-fold slowdowns being very commonly reported). If so, it doesn't help establishing quantitative agreement with experimental results.

Also note that the claim of the authors that their findings might be tested with single molecule experiments is absolutely untrue. Single, or few, molecule strong coupling might just be possible when using optical transitions of very strongly absorbing dyes (this is the work of Jeremy Baumberg with methylene blue, one of the strongest absorbing dyes known). However, molecular vibrations have infrared transitions for which the oscillator strength is at least 2 orders of magnitude higher than those for UV-vis transitions. This basically precludes doing any vibrational strong coupling experiment on individual molecules.

Another simplification is the use of a simple (bi-)harmonic approximation for the reacting molecule and solvent. Anharmonicity is generally very important when discussing vibrational states in organic molecules and coupling between different molecules (in liquids). No rationale was presented to justify this or delimit when and where this might be suitable to use.

Lastly, the authors consider a situation where you have a structured environment. This is really not relevant when discussing chemistry reaction in solution, contrary what the authors state in lines 395-396. Most liquids show molecular movements/re-orientations on a 1-10 ps timescale, which make the environment of molecule in solution highly unstructured and dynamic. It's because of this type of dynamics, a wealth of solution-phase chemistry exists.

2) Comparison with established experimental results.

A part what makes this paper interesting, is to see how far you can go with the simple Hamiltonian proposed here and basically sidestepping the need to multiple molecules or the (unsolvable) multi-nuclei/electron Hamiltonian. However, comparing the trends which the authors predict with those already found experimentally makes me very skeptical of the validity of the model at hand. These are the discrepancies:

1. The authors predict a shift of their response curves, i.e. effect strength as function of cavity detuning, of about 60 cm^{-1} with respect to the energy of the coupled vibration for small interaction strengths. In every published experimental investigation so far, many of which examine very high interaction strengths, the effect maximum is found when tuning exactly to the peak of the absorption spectrum of the vibration. Small deviation (of a few cm^{-1}) might still be possible but not 60 cm^{-1} shifts, which well exceeds the linewidths of most vibrational absorption bands of a typical organic compound.

2. In line with point 1, cavity response curves are predicted which seem much wider than the infrared absorption profiles of a typical vibration. These have typically a full width half maximum of $10\text{-}30 \text{ cm}^{-1}$. Experiments have shown that the cavity detuning response exactly follows the infrared absorption lineshape of the coupled vibration. The authors have not included an absorption spectrum of their (uncoupled) system, which precludes them to verify this correspondence. If they would have and this overlaps with their response curves, it would counter this point of criticism.

3. Experimentally, the strength of coupling to a cavity depends non-linearly on the applied interaction strength/vacuum Rabi splitting: doubling the Rabi splitting will more than double the magnitude of the effect. The main source for this is Thomas et al, Nanophotonics, 9, 2020, 249-255 (which the authors cite). Judging from figure 2 a and figure 4 b and e, this is not the case here. On a sidenote, it is not clear whether or how the given measure of coupling strength η_c corresponds to experimental measures of coupling strength such as fractional Rabi splittings (= Rabi splitting / energy of the bare vibration).

4. The authors predict that, in certain cases, there is both enhancement and suppression of the reaction depending on whether there is positive or negative detuning of the cavity with respect to a certain vibration. Again, the opposite has been observed (see papers below), which is not addressed by the authors.

References for the above are:

- Thomas et al, Angew Chem Intl Ed, 2016 & Thomas et al, Science, 2019
- Lather et al, Angew Chem Intl Ed, 2019
- Hirai et al, Angew Chem Intl Ed, 2020
- Vergauwe et al, Angew Chem Intl Ed, 2019

A more recent work would be Chem10.26434/chemrxiv-2022-wb6vs on ChemRxiv by the group of Blake Simpkins.

In conclusion, I would only consider to accept this manuscript in Nature Communications after some major modifications which address the issues outlined above. My suggestions to help to improve the manuscript:

1. Motivate the construction of the model in detail and cite previous work as much as possible.
2. Include the absorption lineshape of the coupled vibration in order to make the correspondence with the cavity detuning response. The matching of the detuning response to lineshape is the strongest indication experimentalist have currently to show the presence of cavity effects
3. Extend the range of considered coupling strengths. The last results the authors show is a rate suppression if you assume a coupling strength η_c of 0.01 and a large molecule-spectator mode coupling. If you would increase η_c even further, will the observed trend remain? Or will they fall in line with experimental observation? Note that experiments so far have mainly uncovered rate suppressions under relative coupling strengths of at least a few percent (going up to 20 % even when working with water).
4. Consider the effect of simplified H_{solv} Hamiltonians to show the effect of each single type of individual bath effect they include. This would be comparing just to molecule in the cavity to molecule + spectator mode, molecule + solvent, molecule + solvent + solvent spectator mode and, finally, molecule + spectator mode + solvent + solvent spectator mode.
5. Consider an asymmetric potential for the molecule. (Although this might be already beyond the current scope.) A perfectly symmetric potential is actually a very specific case.

Reviewer #3 (Remarks to the Author):

In this article, the authors claim to explain a longstanding open question regarding reaction rates of molecules in a cavity resonant with some vibrational transitions, a branch of research having gained a lot of interest lately after a number of experiments, [1-8] in the article. There have been numerous attempts to explain those findings theoretically, but with varying success, and none which could explain all the effects.

The calculations here are very thorough and the theoretical curves coincide nicely with the experimental ones. This looks very promising. On the other hand, with an ansatz with multiple Hamiltonians and a truly enormous number of adjustable parameters I would be surprised if those curves could not be fitted. As the paper stands now, it is not clear to me whether this is really an explanation or just a good fitting. The authors also emphasize that quantum calculations are necessary. My first impulse on this is "of course" — and some of the prior theory works they authors cite also use a full quantum formalism, thus, I cannot see how the fact that calculations are based on a quantum treatment can be quite as novel and game-changing as the authors seem to claim.

I would strongly suggest to clarify much better what is done - at the moment, any serious information does get lost in "too much stuff." For example, the manuscript would benefit from a table that summarizes the numerical experiments with respect to the various parameters and that compares the numerical experiments to real ones as well as from better foreshadowing in the text of the main results and logical sequence of the figures.

In more detail:

The summary of the paper from the last couple of paragraphs in the introduction are too vague--could they summarize the main takeaways to look out for in the rest of the text, e.g. the dependence of the reaction rate on whether the reactive mode is coupled to a spectator mode, weaker coupling strengths are necessary to observe changes in reaction rate, finite cavity loss is a crucial ingredient? Overall, the specific and interesting insights are scattered throughout the manuscript, and it would be more convenient for the reader to collect them in one place of the manuscript.

Along these lines, it would be helpful to summarize earlier in the paper what studies are done and why. Currently, the structure of the paper is such that experiments are done with parameter X turned off with parameter Y swept and parameter Z sometimes on, and it's difficult to keep up with why one calculation was done and not some other one and how calculation A led to calculation B.

The variables in the expressions for the spectra densities should be labelled and described (the various λ , γ , ω , η)

Since Fig 1 focuses on different limits of H_{solv} , it may be useful to describe structured vs. unstructured environments earlier

Fig 1 is a little confusing at first because the article is presumably about the quantum dynamics of vibrational polariton chemistry, but this is not described until Fig 2, and there is no visual direct comparison between cavity-free and cavity reactions. Confusingly, the authors focus first on the difference between structured and unstructured solvents, which is generally interesting but seems off-topic in this article. Instead, perhaps a more relevant and pressing discussion would be the difference between structured and unstructured molecule-solvent coupling inside of a cavity.

There is generally a lack of mechanistic explanations, especially why quantum calculations give such different results than classical ones. For instance, In Fig 2d, why is the rate constant peaked at the lower polariton energy, and in Fig 2e, why are the resonant frequencies different for the classical vs. quantum cases? In the "Summary" section, it is stated that these differences are due to "quantum light-matter hybridization" and elsewhere, some resonance shifts are due to "renormalization." Both light-matter hybridization and renormalization can also be described classically--what is uniquely "quantum" about this phenomenon that leads to qualitatively different reactivity in a cavity?

There are many variables in this complex model. Could the authors summarize (a) which parameter regimes lead to modifications of chemical reactivity and (b) which parameter regimes real experiments correspond to? As of now, it is challenging to understand whether this theory applies at all to experiments completed thus far, which is fine in my opinion but just needs to be evident

In summary, in its present form, I don't think this manuscript is suitable yet for publication.

Reviewer #1 (Remarks to the Author):

Summary

Comment 0: This work reports on quantum dynamics calculations on a model of a reactive molecule coupled to a cavity mode. The molecule consists of a double-well potential linearly coupled to a dissipative bath, and eventually also linearly coupled to a spectator mode with its own sub-bath. The cavity consists of a single cavity mode linearly coupled (via PF-Hamiltonian) to either the reactive coordinate or to the spectator mode and also features a linear bath modeling cavity losses.

The goal of the study is to provide fully quantum mechanical benchmark results for the modification of thermal chemical reactions in the strong vibrational coupling regime. The rationale for the fully quantum, model-based approach is that up to now studies have been either based on analytical theories (Grote-Hynes and similar) applied to extremely simple Hamiltonians, or based on classical dynamics on realistic potential surfaces.

Key results

1. The modification of the chemical rate occurs sharply at resonances of the molecular system with the cavity mode. Enhancement of chemical rates is related to the stabilization of the products caused by energy transfer to the cavity.
2. In the strong molecule-solvent interaction regime, the cavity can still modify the rate, but in this case, reducing it.
3. It makes a difference whether the cavity couples directly to the reactive coordinate or to the spectator mode. Both enhancement and suppression can take place depending on the specific cavity frequency.

Significance

The paper is in an area of active research, where experimental results on the modification of rates are not yet fully understood, in part because these experiments are performed in complex systems where accurate computational modeling is extremely challenging or simply impossible. Therefore, theoretical contributions that provide new evidence and consolidate existing ideas are important for the advancement of the field.

In this respect, Point 1 above is the conclusion also reached in Ref. 16 on the basis of trajectory-based classical simulations of the chemical rate in an ab-initio potential surface. There, converged reactive-flux calculations find a sharp resonance when the cavity is resonant with a coordinate strongly coupled to the reaction coordinate, similar to the Q_2 coordinate here. Also, Point 2 is in part seen in Ref. 16, although there, the modulation with the cavity frequency was not explored.

Response:

[Note the change in the reference number: Ref. 16 is now Ref. 20. The reference number used below refers to the modified manuscript]

We thank the reviewer for his/her thorough review, and we appreciate the referee's detailed comments, which have helped us clarify our contributions and the importance of the present work.

We partially disagree with the referee's overall assessment of our work. Past work based on classical mechanics not only provides quantitatively different results from those we detail in this work, but, as shown below, *the use of classical mechanics can lead to qualitatively incorrect predictions and provide us with an incorrect understanding of how the cavity radiation mode acts as a dissipator.*

We agree that previous work using classical mechanics has shown cavity frequency dependence of chemical kinetics (e.g. suppression in Ref. 14-16 and enhancement in Ref. 20, 17, and 27). The cavity enhancement of chemical reactivity (that takes place in the energy diffusion-limited regime) observed in both direct classical trajectory calculations (Ref.20) and classical rate-theory (Ref.17) shows *relatively* sharper cavity frequency dependence in the energy diffusion-limited regime compared to other works based on classical mechanics (CM) that operate in the spatial diffusion-limited regime (Ref. 10-12). However, all these studies (including Ref. 20) show a *significantly broader* range of modification of the rate compared to the IR linewidth than our quantum results do, which show widths of the cavity frequency dependence of reaction rates that are comparable to IR linewidths. For example, the linewidths of the rate profiles are ~ 5 times sharper in our work than what is observed in Ref.20 and Ref.17 or the classical calculations carried out in this work. Additionally, the sharp resonances observed in our quantum calculations appear at the transition frequencies between vibrational ground and excited states, whereas classical results show significant (~ 200 cm^{-1}) detuning from these frequencies.

Fig.R1 (now Fig. S5 in the supplementary information): Quantum versus Classical dynamics. Cavity frequency dependent chemical reaction rate obtained from (a) exact quantum dynamics and (b) classical trajectory dynamics at various cavity lifetimes for the same model as Fig.2 in the main text. (c) Maximum rate modification as a function of photon frequency predicted by classical and quantum dynamics with shaded areas indicating 2 standard errors. (d) Cavity frequency dependent chemical rate constant obtained exactly at various cavity couplings. (e) same as d but obtained classically. (f) Maximum rate modification as a function of light-matter coupling.

Importantly, in addition to these quantitative differences, classical mechanics also leads to qualitatively incorrect predictions of the cavity modification of chemical reactivity and provides contrasting physical explanations to those suggested by fully quantum calculations for features that do appear similar:

(1) Classically, the enhancement of chemical reactivity is due to the cavity degree of freedom allowing for faster thermalization (which is the slow step in the energy diffusion-limited regime) inside the **reactant** well, which, within the Pollak-Grabert-Hanggi theory, is captured by the depopulation factor (Ref.17). Within a quantum treatment, coupling to the cavity radiation mode leads to the *splitting* of molecular vibrational levels which enables energy transfer (but not dissipation). In the absence of cavity loss, quantum mechanically the light-matter hybridization itself does not modify solvent interactions (for the “unstructured” spectral density case) and as a result we observe **negligible** modification to chemical reactivity (blue curve in FigR1a), even though we do observe modification of the short to intermediate time nonequilibrium dynamics (as discussed in Supplementary Note 5). On the other hand, classically the *cavity mode itself* provides dissipation and strongly modifies chemical reactivity (blue curve in FigR1b).

When cavity loss is introduced, the loss mechanism allows for faster thermalization of the vibro-polariton excited states in the QM calculations (in comparison to the vibrational excited states) as the far-field modes can be regarded as dissipative solvent modes. This is seen in FigR1a, where this feature is largely absent in FigR1b when using a CM treatment. This discrepancy is further illustrated in FigR1c directly showing the difference between the QM and CM treatment.

Fig.R2: (left) Cavity modified chemical rate obtained quantum mechanically and classically when coupling cavity to the reaction coordinate R. (right) same as left but with cavity coupling to the spectator mode Q.

(2) The CM treatment also shows an incorrect scaling with respect to the light-matter coupling strength. This is illustrated in FigR1d-f. The extent of the modification as a function of the light-matter coupling saturates according to the QM treatment whereas the CM treatment predicts an increasingly large enhancement at large light-matter coupling

strengths (for example at $\eta = 0.005$ the CM treatment makes a $\sim 600\%$ error in estimating the extent of enhancement in Fig.R1f).

(3) The CM treatment breaks down even more when considering a structured spectral density for the solvent. Here, by breakdown we mean that the CM treatment fails to capture the correct cavity frequency dependence, as we notice in all cases considered the rates obtained by CM calculations are well within an order of magnitude of the QM results, but that this is not sufficient to correctly describe cavity modification of chemical rates. We show this in FigR2. In both cases (coupling the cavity to either a spectator mode Q or the reaction coordinate R) the CM treatment predicts **enhancement** of chemical reactivity while the QM treatment shows **suppression** of chemical reactivity. The underlying mechanistic principle of this suppression is that when R or Q is coupled to the cavity, the excited vibrational states of the structured spectral density split due to quantum light-matter hybridization, and this reduces the effective solvent interactions. We find that the classical treatment can not accurately describe this process (see the discussion on the breakdown of CM below)

Fig.R3 : Quantum versus Classical dynamics. Cavity frequency dependent chemical rate obtained from (a) exact quantum dynamics compared with the quantum absorption spectra and (b) same as (a) but obtained classically.

The fundamental reason why the classical treatment breaks down is because it cannot accurately capture vibrational energy transfer for high frequency system modes (also see our response to comment.7 of reviewer 3). Why is this? Work performed in the 1990s points the way. We know that for vibrational energy relaxation in a model system of a harmonic oscillator coupled bilinearly to a harmonic bath, relaxation rates calculated within 2nd order perturbation theory are exact if calculated completely classically (J.S. Bader and B.J. Berne, J. Chem. Phys. 100, 8359 (1994)). The minute non-linearity in the coupling is introduced, this exactness breaks down, more so the higher the frequency of the system is (S.A. Egorov and B.J. Berne, J. Chem. Phys. 107, 6050 (1997)). Discrepancies of a factor of at least 2-3 in the relaxation rate, which could be considered small in the context of the study of vibrational energy relaxation in liquids, are expected. Here in the context of cavity modifications such modifications are considered large. Thus, the differences with fully classical simulation are not necessarily surprising.

These breakdowns should be expected when the Golden Rule is not reliable (as might be the case when the important physics is resonance driven, as we argue here) or when non-linearity is present, or due to both effects. Note that even though our model contains only bilinear coupling between the system and the bath(s), the nonlinearity (anharmonicity) in the reaction coordinate itself will render the conclusions similar. This can be seen by rewriting our model in the eigen-basis of the reaction coordinate Hamiltonian. Now when projected onto this basis, the system-bath coupling term will connect system states that may differ by more than one “quanta” of energy—just as in the non-linearly coupled model of Egorov and Berne cited above. As an illustration of the former (failure of the Golden Rule) issue, consider Fig.R3, where we compare the IR spectrum and the cavity modified chemical rate computed quantum mechanically (Fig.R3a) and classically (Fig.R3b). The width of the classical IR spectrum is very close to that of the quantum IR spectrum, while the difference in the rates is much more dramatic. This difference here is due to the fact that the IR spectrum, which depends on molecule-solvent interactions, is accurately described within the Golden Rule formalism, while the chemical kinetics inside the cavity, where resonances are important, is not.

Changes: To illustrate the qualitative and quantitative disagreements between the QM and CM treatments, we have made the following changes:

- Performed additional CM calculations for the unstructured spectral density model (shown in FigR1) which have been added to the Supplementary Information (Fig.S8) and added the discussion starting with “We next consider the effect of cavity loss on the classical rate profile. As shown in ...”.
- Performed CM calculations for the structured spectral density model which show cavity suppression when using the fully QM method (shown in FigR2). These results have been added in the main text (Fig. 5).
- Performed CM calculations of the IR spectrum and illustrated when the classical treatment works and when it doesn't (chemical kinetics inside the cavity) as shown in FigR3. This has been added to the supporting information.
- Added the following text on **page 1** noting that classical theories break down for cavity-molecule interactions: “Classical treatments of chemical kinetics are often capable of capturing room temperature reaction rates to within an order of magnitude compared to the exact quantum mechanical ones in model calculations [ref]. However, as revealed in this work, in comparison to the exact quantum calculations the classical approaches do not capture the characteristic features of the cavity modified chemical reactivity (namely, the sharp resonant enhancement and suppression of chemical rate) seen in experimental work [ref].”
- Added the text in the on Page 7: “Similar detunings (in the range of \$\sim 150\text{--}200\text{ cm}^{-1}\$ ) of the peak in the classical rate profile compared to the vibrational peaks of the molecular system have been observed in prior classical rate theory calculations [ref], and in molecular dynamics simulations [ref]. In Supplementary Note 4, we compare the classical rate profile to the IR spectrum obtained using classical methods, and observe discrepancies between the peak locations of the two profiles. Interestingly, the width of the classical absorption profile is as sharp as its quantum counterpart shown in Fig.~\ref{fig2}d (also see Fig.S6) in contrast to the width of the chemical rate profile obtained classically. This difference is due to the fact that the IR

spectrum, which depends on molecule-solvent interactions, is accurately described within the Golden Rule formalism, while the chemical kinetics inside the cavity, where resonance effects are important, is not.”

- We have added text in the summary (Pages 10-11) to thoroughly explain the breakdown of classical mechanics starting with “The lessons from the study of vibrational energy relaxation, which is a crucial step in the polaritonic modification of chemical reaction rates, are critical for understanding when we can expect a breakdown of the classical treatment for reaction rates inside a cavity. We expect that there are two potentially simultaneously operative regimes where classical calculations will be in error:
 - When a Golden Rule-level of theory for rates cannot be applied. As shown by Bader and Berne [ref] the Golden Rule vibrational relaxation rate for a purely harmonic, purely classical system is identical to that of a purely harmonic quantum system. However this agreement will break down sharply when one needs to go beyond the Golden Rule to capture the relaxation rates. We expect that this is the case in our system as the rate is governed by resonances, obviating low-order perturbation theory.
 - As shown by Egorov and Berne [ref] when a system is nonlinear, then purely classical theories of vibrational relaxation, while potentially preferable to mixed quantum-classical ones, are not quantitatively accurate even in the Golden Rule limit, especially for high frequency system modes, regardless of the bath frequencies. In the case studied by Egorov and Berne the nonlinearity comes from the form of the system-bath coupling, while in our case it comes from the form of the potential energy curve along the reaction coordinate. It is simple to show that these cases will behave similarly by writing our Hamiltonian in the basis of vibrational eigenfunction of the system. In this basis the system-bath term will couple all eigenstates, and not just ones that differ by one vibrational quantum, just as in the model of Egorov and Berne [ref].”

Comment 1: The significance of the work thence rests very much on the fully quantum nature of the calculations compared to previous studies, although at the cost of introducing a simple model based on one reactive coordinate and linearly coupled oscillators. To better assess this significance, I would like to ask the authors to comment on the following points:

1. I have some problems with statements like in line 114: “These effects emerge from fundamental quantum light-matter interactions and cannot be fully captured with simple classical or semiclassical descriptions of light and matter.”*

Is this really so? Of course, the fundamental and ultimately correct theory to describe matter, light, and light-matter is quantum mechanics, but I do not agree with the implied message: because we are dealing with a hybrid light-matter system, then only quantum mechanics works and can be trusted. If we look into the Hamiltonian, the light-matter coupling terms are of simpler nature and complexity (i.e. lower order of couplings) than the intramolecular potential surfaces of most polyatomic molecular systems. What determines to a large extent how much a CM treatment deviates from QM is the density of quantum states at the total energy of the system. Classical rate calculations including recrossing are extremely successful at capturing chemical rates at room temperature for reactions in the gas phase

and in solution. Missing quantum effects are often due to comparatively small tunneling corrections and zero-point effects. I do not see the cavity making a fundamental difference in this respect. It is a further oscillator at a typical frequency for a vibrational mode.

Hence, the reported differences between the CM and QM description of the rate, and the different widths of the rate resonance, can probably be attributed to the large weight of the tunneling mechanism in this model, with a low and narrow barrier. The tunneling mechanism is more sensitive to resonances, and hence to alterations made by the cavity, than over-the-barrier thermally-activated pathways.

It would be useful to consider this possibility and, if feasible, try to simulate a less tunneling-dominated scenario. The asymmetric potential in Figure S3, but starting from the deep well instead, might suppress deep tunneling and be an interesting test case.

Response: Similar to our response to the previous comment, we do believe that quantum mechanical treatment is indeed necessary to **fully** capture the features of the cavity modification to chemical reaction rates. Of course, classical theories in some limited scenarios can qualitatively provide a correct description of cavity modifications to rates. But, as we illustrated in the series of calculations in response to the previous comment, as well as in the series of calculations presented below, quantum effects are important for capturing the sharp resonant feature of cavity modification. This results from the breakdown of classical treatment beyond the Golden Rule and/or pure harmonic treatment (also see our response to comment.0 above as well as comment.7 to reviewer 3). Thus, we still believe that the statement, "These effects emerge from fundamental quantum light-matter interactions and cannot be fully captured with simple classical or semiclassical descriptions of light and matter." is appropriate. However, we do agree that we need to emphasize the word "fully" in order to avoid any potential misunderstanding. We have made this change in our text.

We note that the frequencies we are dealing with satisfy $\beta\omega \gg 1$ (for vibrational frequency $\omega \sim 1200 \text{ cm}^{-1}$ the first excited state is populated by only 0.3%). We are interested in how the response of the system changes with respect to the frequency of this additional tunable mode, something that requires an accurate description of energy transfer, *which is known to be inaccurately described by classical theories for high frequency modes outside of the golden rule and/or in nonlinear (as defined and discussed in comment.0 above) regimes.*

Note that the very few works that compare exact QM and CM rates [J. Chem. Phys. 101, 7500 (1994)] in condensed phase chemical reaction models with realistic parameters suggest that the accuracy of the classical treatment is within a factor of 2–3 of the exact answer. The modification of the chemical rate that we find here is very much within this range of difference, even though the variation of the differences between the classical and quantum solutions with respect to photon frequency are sharp and striking. Furthermore, the cavity modification to chemical reactivity, where a strongly coupled mode is tuned into and out of resonance with molecular vibrations (something that isn't experienced in typical situations in which classical rate calculations have been benchmarked), is likely to be at the extremes of where classical and quantum mechanical treatments diverge.

Lastly, we thank the reviewer for suggesting the calculation on the asymmetric model to validate our understanding. Our results demonstrate that the sharp features in the chemical rate and the departure from the classical predictions is **not** due to the consideration of models or parameters that are tunneling-dominated. To demonstrate this we carry out the calculation suggested by the reviewer, in addition to investigating several more scenarios where the barrier is much higher and broader. We find that our conclusions are general and the basic principle and features that we have obtained in the original manuscript remain the same.

First, we consider the scenario where the barrier is increased to 3000 cm^{-1} such that at least two vibrational levels remain well below the barrier. The overall results we find are qualitatively the same as previously seen, with classical mechanical results showing a much broader and shifted rate profile compared to its quantum counterpart.

Fig.R4: Results for the case of a 3000 cm^{-1} barrier. (a) Potential energy surface and vibrational eigenstates. (b) Potential energy surface and photon-dressed vibrational states (with 0 and 1 photons). (c) vibro-polariton states as a function of photon frequency. (d) Rate constant as a function of solvent friction. (e) Cavity modified chemical rate constant for various cavity lifetimes. (f) Cavity modified rate profile obtained quantum mechanically and classically.

We go one step further and compute the cavity modification to chemical reaction rate when the barrier is 4000 cm^{-1} such that three vibrational levels remain well below the barrier. The results are again similar to the 3000 cm^{-1} case.

Fig.R5: Results with 4000 cm⁻¹ barrier. (a) Potential energy surface and vibrational eigenstates. (b) Chemical rate profile at various solvent friction. (c) chemical rate constant obtained classically (green) vs quantum mechanically (blue).

Finally we carry out the calculation suggested by the reviewer, namely the backward rate for the asymmetric model, where we find that the structure of the cavity modified backward rate is visually identical to the forward rate (while the magnitude of the chemical rate is different by two orders of magnitude). This is because the ratio of the forward and the backward rate is equal to the ratio of the equilibrium populations in the reactant and product wells which we find to be insensitive (to less than 0.1% of its value) to alteration of the cavity photon frequency.

Fig.R6: Results with asymmetric barrier. (a) Potential energy surface and vibrational eigenstates. (b) Forward and (c) backward chemical rate profile at two cavity lifetimes.

Finally we also point out that the modifications to (ground vibrational state) tunneling-dominated reactions (such as studied in FigS2 or the off-resonant suppression in Fig.3c) do not show strong cavity frequency dependence, which can be understood by using a polaron transformation (see details in J. Phys. Chem. Lett. 11, 9215 (2020)) as explained in the main text starting with “Here, we also observe an off-resonant suppression of the chemical rate at higher light-matter couplings. This effect can be understood by applying a polaron transformation, leading to a rescaling of ...”.

Changes: We have performed several additional calculations (including the one suggested by the reviewer) which should convince the reviewer of the importance of the quantum effects in cavity modification to chemical reactivity. These results as a whole demonstrate that vibration-cavity energy transfer as well as how the cavity modifies molecule-solvent interactions quantum mechanically are crucial for understanding the sharp features of the cavity modified chemical rate. We list our changes below:

- We perform calculations with higher barriers : (i) 3000 cm⁻¹ and (ii) 4000 cm⁻¹ and have added them to the Supporting Information (Fig.S10 and Fig.S11).
- We compute the backward rate for the asymmetric model as suggested by the model and have added these results in the Supporting Information (Fig.S3)

Comment 2: How realistic is a model based on linear couplings only? Although exact HEOM calculations are possible on it, it may oversimplify other aspects of the dynamics. For example, its dynamics may be much less chaotic than those of realistic potentials with

higher-order anharmonic couplings. This might explain the lack of an effect in the absence of cavity losses (Fig. 3b, text in lines 345 and below).

Response: The model based on linear coupling to a bath is the standard one for the following reason: In the classical limit this model produces *exactly* the Kramers model of a chemical reaction in the presence of memory friction. This means that the model is much more general than might be expected by its harmonic bath form (see The Journal of chemical physics 85 (2), 865-867 (1986)). Indeed, models of this form may be used even for reactions in fluids (which are anharmonic!) where the spectral density of the effective harmonic bath is computed from 2-point simulations of the bath response (see for example (J. Chem. Phys. 110, 5307 (1999))). A model with *non-linear* coupling between the system and the bath would imply memory friction that is explicitly dependent on the reaction coordinate (so called “reaction coordinate-dependent” friction). It remains unclear how important such effects are. In extreme cases (ion pair dissociation in a fluid) simulations show some dependence of the friction kernel on the reaction coordinate (see: J. Chem. Phys. 93, 6804 (1990)) although these effects are not very large. Further, for vibrational energy relaxation in a fluid these effects appear to be almost completely absent (see: J. Chem. Phys. 98, 7301 (1993)). Lastly, there is some theoretical evidence that even when reaction coordinate-dependent friction is present, that the effects on the rate are negligible (see: J. Chem. Phys. 103, 10176 (1995)). Thus, overall we believe that the model we have considered should be a good one for capturing the behavior of chemical reactions in liquids.

We have also considered the case of non-linear forms of the solute dipole (gaussian multiplied by linear-dipole moments) and have added these results to the SI. Here, we observed that the conclusions concerning cavity losses do not change. Note that (as explained in detail in response to comment 1, see above) the role of cavity loss has little to do with the shape of the molecular dipole.

Fig.R7. Results with non-linear dipole. (a) Potential energy surface and vibrational eigenstates. (b) cavity modified chemical rate profile at various light-matter coupling and (c) at various cavity lifetimes.

Changes: We have performed calculations considering nonlinear interactions between cavity and molecule. We have added these calculations to the supporting information (Fig.S10) and added the text in the supplementary information starting with “In the main text we have considered the case of bi-linear coupling between the molecular system and cavity mode. In doing so we have made an assumption of a linear dipole moment for the system, here we consider lifting this assumption and consider a non-linear dipole operator for the molecular system of the form...”

Comment 3: In lines 571-574 one reads the statement that the enhancement or suppression of chemical rates near resonance with molecular vibrations had not been captured by previous theoretical studies. This finding has been reported on the basis of classical flux calculations before. In the present contribution, the authors report this resonant behavior also in the strong solvent friction regime, which had been seen as well, but not studied as a function of the cavity frequency. Maybe these lines could be rewritten to more accurately reflect the current state of understanding.

Response: We agree that cavity frequency dependence of chemical reactivity has been seen within classical treatment which includes Ref. 10-13, 16, and 23 and we did acknowledge that in various places including in the introduction. What we intended to state is that a sharp feature in the cavity modified chemical rate (with peaks appearing near vibrational transitions— i.e. resonance) that is of the same order of magnitude as the absorption profile has been seen for the first time in this work. We have also clearly demonstrated that classical flux calculations cannot capture (even qualitatively, such as Fig.R2 or modified Fig.5) the sharp/resonant features obtained in our work.

We have modified the text to clarify that previous works based on classical theories have predicted both enhancement and suppression and that our work is different in that we capture much sharper features and make predictions that cannot be made with classical theories.

Changes: We have added the sentence: “Previous work that treats the molecular and radiation degrees of freedom classically have found that cavity modification of chemical kinetics is photon frequency dependent and maximum suppression occur when the photon frequency is close to the barrier frequency [ref] in the spatial diffusion-limited regime (exhibiting suppression) or when photon frequency is close to well frequency in the energy diffusion-limited regime [ref]. However, in these works the rate profile is much broader ($\sim 500\text{-}5000\text{ cm}^{-1}$) than the corresponding absorption profiles of the bare molecular system and the peak modification typically appeared comparatively far from the relevant vibrational transition frequencies.”

Comment 4: Is the formation of the Rabi splitting in the absorption spectrum really determining the cavity effect as shown in Fig. 2d and discussed in lines 262 and below? On the one hand, the formation of polaritonic resonances can be seen in fully classical treatments [e.g. Tao et. al, Angew. Chem. 60, 15533 (2021)]. On the other hand, this viewpoint is not adopted when considering the coupling of R with Q , although exactly the same consideration could have been made.

Response: First, there seems to be a confusion with respect to the reviewer’s point in that we do not assert that “the formation of the Rabi splitting in the absorption spectrum really determining the cavity effect” in line 262 and below (the text only explains why Rabi-splitting is observed using simple JC Hamiltonian). The Rabi-splitting is *indicative* of a strong light-matter interaction which is related to the splitting of vibrational energy levels. Consequently, splitting of vibrational energy levels leads to the modification of chemical rate in all model systems including when considering the coupling of R with Q . Note that coupling R and Q also creates a splitting (similar to the light-matter coupling) and if we scan

the chemical rate as a function of the Q mode frequency (an unrealistic thing to do, given that we cannot typically modify matter vibrations at will as we can for the cavity) we will obtain similar results as to what we find when scanning the the cavity mode frequency. This is consistent with the quantum mechanical modification by cavity coupling discussed in this work.

Overall, the Rabi-splitting is related to the modification of chemical reactivity in that both originate from strong light-matter coupling. We maintain the viewpoint that near resonance the strong light-matter interactions modifies the vibrational energy relaxation (which can lead to altered dissipation)of the molecular subsystem, and we have used this viewpoint in all cases. We have modified the text to clarify these and to de-emphasize the connection between Rabi-splitting and chemical rate.

Second, it is true that one can obtain reasonable polariton absorption using classical mechanics. The reason why the classical treatment works is because absorption spectra are well estimated within the linear response regime (emerging from weakly coupled fields coupling to polariton hybrid systems), and are dominated by the dynamics arising near the bottom of the reactant or product wells (which are well described within a purely harmonic model). But this is different from the question of modifying chemical reactivity, where we are interested in seeing how strongly coupled cavity fields modify the dynamics of the molecular subsystem (as it transitions between well and barrier regions - and experiences strongly anharmonic interactions) - which strongly depends on the details of energy transfer beyond this regime (also see our thorough response to Comment.0 of reviewer 1). Therefore when we are interested in cavity modified chemical kinetics a quantum treatment is necessary. We have added a few sentences to clarify this.

Changes:

- Added the sentence (Page 9) “Note that for $\mathcal{C}_Q = 5 \times 10^{-6}$ we also see a splitting of the rate profile with peaks corresponding to the normal modes formed by strongly coupling the R and Q modes (a Rabi-splitting due to coupling between two nearly resonant vibrational modes).”
- We have removed the sentence “~~Importantly, the resonant structure of the rate profile appears due to the modification of the environmental friction which originates from the Rabi-splitting caused by quantum light-matter interactions.~~” to de-emphasize the connection between Rabi-splitting.
- We have added the sentences on Page 7: “Interestingly, the width of the classical absorption profile is as sharp as its quantum counterpart shown in Fig.~\ref{fig2}d (also see Fig.S6) in contrast to the width of the chemical rate profile obtained classically. This difference is due to the fact that the IR spectrum, which depends on molecule-solvent interactions, is accurately described within the Golden Rule formalism, while the chemical kinetics inside the cavity, where resonance effects are important, is not.”

Minor points --

Comment 5: What is the temperature in the quantum dynamical HEOM simulations? I could not identify this parameter in the main article or the methods section. The supporting information reports 300 K, but only for the classical case.

Response and Change: We thank the reviewer for pointing this out. All calculations were done at 300K. We have mentioned this in page 5 when introducing β .

Comment 6: Line 202: Shouldn't Fig 1b be Fig 1c?

Response and Change: We thank the reviewer for pointing this out, the text has been updated to reflect this.

Summary: In my opinion, this is an interesting contribution with insightful elements. Its technical quality and validity are without doubt of a high standard. I am not yet fully convinced that its significance matches the Nat. Comm. criteria. Its main conclusions solidify and partially extend previous theoretical and computational observations. It builds strongly upon the notion that quantum effects are fundamental in the vibrational strong coupling. The way this is argued does not fully convince me, although this can maybe be remedied.

Response: We again thank the reviewer for his/her insightful comments and suggestions. We still maintain that the full treatment of quantum effects is very important for solving the mysteries of VSC, and we demonstrate the inability of the classical treatment (through a new set of results : see Fig. S8, Fig. S11f, Fig. S12c, modified Fig.2e and Fig5) to capture most of the features exposed in this work. We do not believe that our work merely extends previous theoretical and computational studies. The sharp resonant suppression and enhancement (that matches the shape of absorption spectra) shown in our work has not been observed before, which is a central feature of VSC experiments. Previous work (including the references cited by the reviewer) show much broader and often off-resonant modification (where peak cavity chemical rate modification occurs far from any peaks in the absorption spectra) of chemical reactivity. The aspect of tuning the solvent-molecule interaction (when including a spectator mode) in or out of resonance, a model that is pertinent to the existing VSC experiments, has also been theoretically demonstrated here for the first time. Overall, we believe that the physical insights of the cavity modification we provide in this work through exact quantum dynamics are general and are unlike any previous studies. Importantly, many of these insights cannot be drawn from classical simulations. We hope that the new set of quantum and classical simulations, coupled with the clarifications made here in the text, convince the reviewer that our work is worthy of publication.

Reviewer #2 (Remarks to the Author):

Comment: The authors present a theoretical model for chemical reactions under vibrational strong coupling (VSC). This means a molecular vibration has been entangled to the confined electromagnetic field of a photonic cavity, resulting in the creation of vibrational light-matter states, a.k.a. (vibro-)polaritons. This is an exciting new field on the border of physics and chemistry as it promises to develop an entirely new approach for controlling chemical reactions. The big challenge the field is now facing, is to understand the mechanism behind the observed modifications of reactions executed in a resonant cavity.

I am active in the field as an experimentalist for many years now, in which I consider myself an expert. Therefore, I will mainly focus on the value of the proposed model to explain VCS chemistry experiments. The implementation of the computational approach needs to be verified by another expert.

To start with the positive, the authors bring forward an interesting attempt for a fully quantum-mechanical model for chemistry under VSC and have succeeded in calculating a wealth of characteristics. Without a doubt, this is testament to their strength as a theoretical group. The focus they put on including dissipative interactions with spectator vibrations and the solvent bath is interesting and constitutes an important, original contribution to the field. Several trends they observe, are counter-intuitive but fascinating. Their overall explanation for VSC effects based on considering VSC modification of the interaction of the molecule with its environment seems like a strong candidate to answer the outstanding question in the field. The manuscript is also quite clearly written (although there are still a few odd sentences to be found, likes lines 275-278 and line 518-521).

Response and Change: We thank the reviewer for his/her encouraging comments. We have fixed the sentences that did not read well, as pointed out by the reviewer.

Comment 1. However, I have concerns from 2 different perspectives which make me doubt the value of the proposed model for polaritonic chemistry.

Modelling of a reaction: In order to use an approach which allows the authors to make their quantum mechanical model exactly solvable some serious simplifications were used in constructing their Hamiltonian, esp. regarding modeling of the reacting molecule. These choices are not motivated in the text, which is a serious weakness. Citing previous work which uses such simplifications and obtain useful results would heavily strengthen the case the authors want to make.

The most serious simplification is only considering a single reacting molecule. In strong coupling this will not give you the polaritonic dark states. These are generally considered to have a major influence as the Jaynes-Cumming-Tavis model predicts that there should be always so many of them: $N-1$ dark states when N molecules are coupled. In vibrational strong coupling, you couple typical 10^5 - 10^6 dipoles/molecules into a set of coherent polaritons states. Many theoreticians in the field of organic polaritonics now consider the many molecules case needs to be considered in order to successfully explain and predict effects on chemical reaction and other material properties. This approximation might also

explain why only relatively small relative effects are found here (< 30 %), while much larger effects have been measured (4-to-5-fold slowdowns being very commonly reported). If so, it doesn't help establishing quantitative agreement with experimental results.

Response: We have chosen a model and parameters that contain standard ingredients (e.g., a reaction coordinate profile typical of realistic reactions in liquid solvents, a Hamiltonian that produces the standard model of memory friction on the reaction coordinate, and a coupling between the system and the cavity of the standard microscopic form, all with reasonable parameters) of the coupling of molecular, solvent and cavity degrees of freedom. At the same time we fully agree that the model used in this work does not directly address the issue of collectivity, and we acknowledged this in our original text. We have added some sentences in the introduction as well to clarify this limitation.

That said, this does not mean our results are irrelevant for understanding collective experiments. First, we note that while the referee suggests the results we find show <30% alterations in reaction rates, our original figure Fig.3 shows changes approaching a factor of 2 in the rate depending on the rate of cavity loss. These results, while not large compared to some of the biggest experimental claims, are not far from some of the effects experimentally reported in the literature, *and are substantially larger than any previous theoretical reports for single molecule calculations.* Furthermore the nature of the enhancement and suppression is qualitatively far closer to experimentally reported results than previous calculations in terms of the sharpness and the spectral locations of the features, as we will highlight further below. These facts suggest that while our calculations do not include the potentially important collective effects noted by the referee, *any future work on collective effects likely needs to consider the quantum nature of the problem as we have, and that some aspects of what we expose will undoubtedly survive in the collective regime as well.*

In a more direct and quantitative sense, we point out that the model system we have studied that includes a spectator mode Q coupled to a cavity reduces to an idealized model for including collective effects (J. Chem. Phys. 156, 014101 (2022)), where Q can be viewed as a collective solvent coordinate such that the light-matter coupling scales with \sqrt{N} with N as the number of solvent DOF *directly* coupled to the molecular system. While beyond the scope of the current paper, our current unpublished results suggest that for reasonable parameter values such a model indeed shows substantial rate modifications in the collective limit that are intimately connected to the physical behaviors presented here.

Our results show that a set of far-field dissipative modes (inducing loss) that couple to the cavity mode strongly influence the chemical reactivity of a molecular system. When considering an ensemble of molecular vibrations, from the perspective of one reactive molecule, the coupling of the rest of the molecules to the cavity is structurally similar to how far field dissipative modes couples to the cavity. Therefore, in the collective regime, collective coupling of all molecular vibrations to the cavity may lead to additional dissipation (playing a similar role to cavity loss), but to what extent remains an open question. We are actively working in this direction. Thus, overall, we strongly believe that the present work provides us with a solid foundation to potentially resolve the mysteries of VSC mediated chemistry in the collective regime as well. We emphasize again that the main result of our work, *namely the necessity of a full treatment of quantum effects*, resides on general principles of resonance and energy relaxation that would still be present in the collective

case. Thus, our results in this regard are not confined to the single molecule case, and any future treatment of collective effects should incorporate them.

Changes:

- We have added some text in the introduction to clarify the nature of our work, starting with “To this end, the work presented here does not directly address the issue of collective effects. We would like to point out, however, that some idealized models of collective polaritonic behavior [ref] reduce to models similar to those considered below (namely a reaction coordinate strongly coupled to a spectator/collective solvent mode), and further that single molecule studies are possible in principle [ref], although it may be technically challenging to do so as the oscillator strengths for vibrations are typically several orders of magnitude smaller than electronic transitions.”
- On page 8 we have added a few sentences to clarify the similarity of the model employed in our work and previous work studying collectivity, starting with “Note that while in the present work we do not directly address the issue of collectivity, the model presented here is structurally the same as an $\{it\}$ idealized model for studying collective effects [ref]. That is, the mode Q can be viewed as a collective solvent coordinate, such that the light-matter coupling η_c scales with \sqrt{N} where N is the number of solvent modes directly coupling to the molecule.”
- On page 7 we have added a few sentences to point out the possible connections to collective effects: “This result also opens up interesting questions regarding the collective cavity modification of chemical reactivity. When considering N non-interacting molecules [ref], from the perspective of one reactive molecule, the coupling of the rest of the molecules to the radiation mode is structurally the same as the far field modes that couple to the cavity mode describing cavity loss. Therefore, the collective coupling may provide an additional source of dissipation (playing a similar role to cavity loss) and may modify chemical reactivity, but to what extent remains an open question reserved for future study.”

Comment 2. Also note that the claim of the authors that their findings might be tested with single molecule experiments is absolutely untrue. Single, or few, molecule strong coupling might just be possible when using optical transitions of very strongly absorbing dyes (this is the work of Jeremy Baumberg with methylene blue, one of the strongest absorbing dyes known). However, molecular vibrations have infrared transitions for which the oscillator strength is at least 2 orders of magnitude higher than those for UV-vis transitions. This basically precludes doing any vibrational strong coupling experiment on individual molecules.

Response: We fully agree with the referee that such experiments are extremely challenging, but regardless of the difficulty, the single molecule limit is most certainly not unphysical. We respectfully disagree in that it is “absolutely untrue” for a single molecule experiment to exist in the IR regime. For example, a single molecule plasmonic setup can be extended to the IR regime, as has been shown by Nitzan et. al. (J. Phys. Chem. Lett. 2022 13 (41), 9673-9678). Parameters such as the quantization volume, cavity photon frequency, etc. estimated in this

work are well within the range used in our work. A recent experiment also points to the possibility of achieving VSC with few molecules (J. Phys. Chem. Lett. 2021, 12, 3171–3175) which states “*The most remarkable feature of our system is that the VSC occurs on a single nanogap patch antenna level and the possibility of further reduction of mode volume to achieve VSC with just a few tens of molecules.*” Also note that while the oscillator strength is at least 2 orders of magnitude lower in the IR regime, the couplings/Rabi-splittings required for modifying chemical reactivity in the IR regime are also smaller by a few orders of magnitude. We have now cited these papers in our main text and pointed out the difficulties in achieving strong cavity coupling in the IR regime.

Change:

- We have added a sentence on Page 7 : “The parameters used here lie within the range of parameters theoretically estimated for single-molecule plasmonic cavities in the IR regime [ref] consistent with recent experimental work [ref].”
- We have modified the text in the introduction which reads as “... although it may be technically challenging to do so as the oscillator strengths for vibrations are typically several orders of magnitude smaller than electronic transitions.”
- We have added a sentence in the introduction “The parameters used in this work are well within the range of theoretically estimated and experimentally achievable parameters for plasmonic cavities in the single molecule limit [ref].”

Comment 3. Another simplification is the use of a simple (bi-)harmonic approximation for the reacting molecule and solvent. Anharmonicity is generally very important when discussing vibrational states in organic molecules and coupling between different molecules (in liquids). No rationale was presented to justify this or delimit when and where this might be suitable to use.

Response: The referee's comment is an important one that needs to be clarified. Indeed, it would appear that our model system is disconnected from reactions that occur in real liquids which are definitely not harmonic! However, as is well known in the theoretical community (but perhaps less appreciated in the experimental community), *our model is far more general than its harmonic form and is indeed known to be a good model for chemical reactions in liquids.*

The important insight is the fact that a bilinearly coupled harmonic bath *exactly* captures the physics of *any system* that is in the linear response regime. Such systems represent most liquids. In fact this is why the harmonic free energy picture behind Marcus theory works quantitatively in liquids such as water-while the system is very far from harmonic, the central limit theorem renders the system linearly responding. Hence computer simulations show that the exact free energy profiles for electron transfer reactions in water are nearly perfectly harmonic! The referee should, for example, see the seminal simulations of such a process by Chandler and coworkers (J. Chem. Phys. 89, 3248 (1988)).

More specifically, concerning chemical reaction rate theory, Zwanzig used this insight to show that the Kramers reaction rate problem in the presence of memory friction-which is *the*

canonical model for chemical reactions in liquids (see: Rev. Mod. Phys. 62, 251 (1990)) is *exactly* one-to-one mappable onto our Hamiltonian (see: J. Stat. Phys. 9, 3, 215, (1973)). Thus, with an appropriately chosen coupling value and distribution of harmonic frequencies, our model exactly will reproduce a model of a reaction coordinate in a liquid damped by memory friction with noise that forms the basis of nearly all models of chemical reactions in liquids. As a consequence, one may perform a molecular dynamics simulation of a liquid, numerically compute the friction kernel, infer the spectrum of the fluctuating force, and then parametrize a Hamiltonian of *exactly* the form used in our work that will reproduce all of the features and rates that would have been obtained directly from the molecular dynamics simulations of the liquid.

In this regard, a linearly responding anharmonic liquid will always have a proxy Hamiltonian of the harmonic form we use, and in fact this mapping has been used countless times to understand reactions in liquids. For example, the well-established Pollak-Grabert-Hänggi theory (J. Chem. Phys. 91, 4073 (1989)) that has been successfully employed for understanding chemical reaction rates in anharmonic condensed phases such as liquids (see Rev. Mod. Phys. 62, 251 (1990)) uses our form of the Hamiltonian. These facts are nicely summarized by Marcus and coworkers in (J. Chem. Phys. 110, 5307 (1999)). *Thus the model we employ is far more general and valid for anharmonic systems like liquids than its harmonic form implies.* We have clarified this in the main text.

Change: We have added the following text when introducing our model on Page 5: “This model of a bilinearly coupled harmonic solvent is known to capture solvent mediated dissipative processes [ref] and is central to many rate theories in the liquid-state [ref] as well as numerical treatments of chemical reaction processes that include dissipative effects due to condensed phase environmental degrees of freedom [ref]. For regimes in which a linear response treatment of the solvent degrees of freedom is valid, an anharmonic solvent can rigorously be mapped onto such harmonic bath model [ref].”

Comment 4. Lastly, the authors consider a situation where you have a structured environment. This is really not relevant when discussing chemistry reaction in solution, contrary what the authors state is lines 395-396. Most liquids show molecular movements/re-orientations on a 1-10 ps timescale, which make the environment of molecule in solution highly unstructured and dynamic. It's because of this type of dynamics, a wealth of solution-phase chemistry exists.

Response: This is a misunderstanding of what is meant by a “structured environment” in our work and we apologize for the misunderstanding. Here “structured” refers only to the spectral density which describes how the solvent interacts with the reacting system and makes no assumption concerning the **spatial structure** of the solvent. In this context “structured” simply means that the weighted density of states of the solvent bath has some sharp spectral features, as would be true in a realistic situation where the molecule has vibrational modes that are not collinear with the reaction coordinate or the solvent has prominent vibrational modes. This form of a spectral density is often used for modeling solvent interactions in the condensed phase (J. Chem. Phys. 90, 3537 (1989), J. Chem. Phys. 110, 936 (1999), J. Chem. Phys. 110, 465 (1999), J. Chem. Phys. 142, 244110 (2015), J. Chem. Phys. 114, 2910 (2001), Physics Reports, 168, 3, 115-207 (1988), J. Chem. Phys.

135.064504 (2011)), and here the use of the term “structured” is as we have used it, although we agree that this may cause confusion. We have added some clarification in the main text.

Change:

- We have added the text on Page 5: “... in the following simulations we will consider two specific cases for the solvent. In the first, we will consider no coupling between the system and spectator mode, i.e. $J_{CQ} = 0$. In this case, the spectral density for the solvent is described by the broad, unstructured spectral density $J_U(\omega)$, and as such will be referred to as the “unstructured” environment. In the second, we will consider non-zero coupling between the system and spectator mode. In this case, applying a normal mode transformation to the solvent Hamiltonian, we find that the presence of the spectator mode gives rise to a sharp peak in the solvent spectral density $J_S(\omega)$, as shown in Fig.~\ref{fig1}b). Note that here we use the term “structured” to refer to the sharp structure in the spectral density at specified frequencies which does not imply positional or orientational structure in the environment.”

Comment 5. Comparison with established experimental results.

A part what makes this paper interesting, is to see how far you can go with the simple Hamiltonian proposed here and basically sidestepping the need to multiple molecules or the (unsolvable) multi-nuclei/electron Hamiltonian. However, comparing the trends which the authors predict with those already found experimentally makes me very skeptical of the validity of the model at hand.

Response. We must clarify that our goal is not to directly reproduce (nor do we claim to reproduce) existing experiments, and a direct comparison to existing experiments cannot be made as we are operating in the single molecule limit while the experiments operate in the collective regime. Our goal has been to explore and understand if and how a cavity could modify single molecule chemical reactivity, how physical properties of the molecular system play a role, and to what degree a fully quantum treatment of the problem is necessary. To this end, we have studied a series of related models and scanned relevant parameters such as photon frequency, light-matter coupling strength, solvents friction, cavity loss rate etc. to understand how each individual parameter plays a role in answering these questions. We believe it is not necessary for our results to match the present experiments, and indeed we know of no microscopic theories that currently do.

However we do find it interesting and perhaps suggestive that even when disregarding the issue of collective effects, we do capture *some* important qualitative trends surprisingly well when compared to the observed experiments. Certainly our results come closer than previous microscopic theoretical works does in this regard. Below is a comparison of an enhancement and a suppression of chemical reactivity in our work and previous experiments. In (a) we consider a reaction coordinate coupled to a dissipative bath in the energy diffusion-limited regime for which the reactivity is enhanced. On the other hand, in (b) we consider a case where the reaction coordinate is strongly coupled to a spectator mode,

as is the case in (d). Note the splitting in the transmission spectra (corresponding to the out-of-cavity experiment) in (d) which may indicate the hybridization of the different C-Si vibrations, is similar to (b). Our theoretical result, as in (d), predicts suppression, albeit weaker than seen in the experiment. Note that under inhomogeneous broadening (which does not exist for a single molecule) the two peaks of the absorption will merge and will have a similar lineshape as the transmission spectra in (d).

Lather et. al. Angew. Chem. Int. Ed. 2019, 58, 10635–10638

Thomas et. al. Angew. Chem. Int. Ed. 2016, 55, 11462–11466

Fig.R8. Cavity modification of chemical reaction rate in (a) an unstructured and (b) a structured (including a spectator mode) bath compared to absorption spectra. Qualitatively similar results seen in experiments showing (c) enhancement and (d) suppression and a signature of line splitting (d).

No Changes are made.

Comment 6. These are the discrepancies: 1. The authors predict a shift of their response curves, i.e. effect strength as function of cavity detuning, of about 60 cm^{-1} with respect to the energy of the coupled vibration for small interaction strengths. In every published experimental investigation so far, many of which examine very high interaction strengths, the effect maximum is found when tuning exactly to the peak of the absorption spectrum of the vibration. Small deviation (of a few cm^{-1}) might still be possible but not 60 cm^{-1} shifts, which well exceeds the linewidths of most vibrational absorption bands of a typical organic compound.

Response. We think that the reviewer has misunderstood some aspects of our presented results. We observe a 60 cm^{-1} shift of the rate profile in a few specific scenarios while in other setups we observe resonant modifications. *Therefore this shift is not a global trend that we observe.* In fact in all the scenarios we have considered (such as the cases presented in Fig2, Fig.3, Fig.4e, Fig.5), we find that generally at weaker light-matter coupling we observe *resonant* modification. In a few cases (such as in Fig.4b, Fig.4c, Fig.4f) we observe some shift when increasing the light-matter coupling. An example is shown below in Fig.R9 where for $\eta_c = 0.00125$ the peak cavity modification appears near the peak of the absorption, and with increasing η_c it red shifts.

Overall, we cannot directly compare our results to experiments, and even if we were to do so we note that we observe resonant modification in most parameter regimes.

Fig.R9. Comparison absorption spectra with cavity modified rate profile.

As a side note, while we agree with the reviewer that in most VSC experiments it has been observed that the peak chemical kinetics modification occurs near some vibration frequency, in (J. Chem. Phys. 155, 241103 (2021), see Fig. 5) the authors found some enhancement of $\sim 60\%$ at 50 cm^{-1} detuning with a 20% standard deviation. In another example, see the figure to the right from (Thomas et. al. Science 363, 615, 2019), where the rate profile matches some vibrational peaks but also shows strong modulations away from any existing IR peak, such as near $\sim 1175\text{ cm}^{-1}$.

Change: No changes are made.

Comment 7. In line with point 1, cavity response curves are predicted which seem much wider than the infrared absorption profiles of a typical vibration. These have typically a full width half maximum of $10\text{-}30\text{ cm}^{-1}$. Experiments have shown that the cavity detuning response exactly follows the infrared absorption lineshape of the coupled vibration. The authors have not included an absorption spectrum of their (uncoupled) system, which precludes them to verify this correspondence. If they would have and this overlaps with their response curves, it would counter this point of criticism.

Response. We apologize for not making it clear that the absorption spectra has a similar linewidth as our rate profile. We have modified the original Fig.2 to clarify this. Here we present the comparison for the case of an “unstructured” spectral density within the meaning explained above. Also note the failure of the classical calculations to capture this qualitative trend. Note that the width of the cavity rate profile depends on light-matter coupling and cavity loss, so depending on the parameters we find a rate profile with a width typically less than twice the absorption line shape.

Changes. We have replaced Fig. 2d with the figure (left panel) above, adding the absorption spectra of the uncoupled molecule. We have added additional absorption spectra in the supporting information.

Comment 8. Experimentally, the strength of coupling to a cavity depends non-linearly on the applied interaction strength/vacuum Rabi splitting: doubling the Rabi splitting will more than double the magnitude of the effect. The main source for this is Thomas et al, Nanophotonics, 9, 2020, 249-255 (which the authors cite). Judging from figure 2 a and figure 4 b and e, this is not the case here. On a sidenote, it is not clear whether or how the given measure of coupling strength ηc corresponds to experimental measures of coupling strength such as fractional Rabi splittings (= Rabi splitting / energy of the bare vibration).

Response. We believe there has been some confusion on the part of the reviewer and disagree that our results qualitatively deviate from the trend observed in (Nanophotonics, 9, 2020, 249-255). We find that the cavity modification follows a non-linear relationship with coupling strength as shown here (in the right), such that the effect of cavity saturates at some high cavity coupling. We also observe a non-linear effect in the case where we observe suppression (see Fig. R11). The **same** trend is seen in Fig.3D in (Nanophotonics, 9, 2020, 249-255) where the cavity modification reaches a plateau for a Rabi-splitting larger than 90 cm^{-1} , which seems contrary to what the reviewer suggests, namely that doubling the Rabi splitting will more than double the magnitude of the effect.

Fig.R11 Comparison between (left) this work (right) experiment. Note we operate in the **single molecule** limit—unlike the experiment which operates in the collective regime.

We apologize for not making it clear how the normalized light-matter coupling relates to the η_c used in this work. We have added some text to clarify this. All results shown in this work lie below the ultra-strong coupling regime (ultra-strong coupling refers to light-matter couplings for which Rabi-splitting/ $2\omega_c > 0.1$), such that fractional Rabi splittings are < 0.1 (in figure 2 it is 0.0125).

Change:

- We have added text on Page 6 to clarify the relationship between η_c and the normalized light-matter coupling (fractional Rabi splittings), starting with “the normalized light-matter coupling $\bar{\eta} = \frac{\Omega_{\text{c}}}{2\omega_{\text{c}}}$ ”, which places the light-matter coupling in the weak-strong ($\bar{\eta} < 0.1$), ultra-strong ($1.0 > \bar{\eta} > 0.1$) and deep-strong ($\bar{\eta} > 1$) coupling regimes, is approximately related to η_{c} as $\bar{\eta} \approx \eta_{\text{c}}/\sqrt{2\omega_0}$ when approximating the reactant well as harmonic and using the Jaynes–Cummings model. All results presented here lie in the weak-strong coupling regime and the results presented in Fig. 2 use $\bar{\eta} \approx 0.0125$.”

Comment 9. The authors predict that, in certain cases, there is both enhancement and suppression of the reaction depending on whether there is positive or negative detuning of the cavity with respect to a certain vibration. Again, the opposite has been observed (see papers below), which is not addressed by the authors.

References for the above are:

- Thomas et al, Angew Chem Intl Ed, 2016 & Thomas et al, Science, 2019
- Lather et al, Angew Chem Intl Ed, 2019
- Hirai et al, Angew Chem Intl Ed, 2020
- Vergauwe et al, Angew Chem Intl Ed, 2019

A more recent work would be Chem10.26434/chemrxiv-2022-wb6vs on ChemRxiv by the group of Blake Simpkins.

Response. Again, we have attempted to scan all relevant parameter regimes in our model systems, and explored various possibilities of cavity modification to gain a complete understanding of how the cavity modifies chemical rates. We have attempted to make a thorough investigation of what is possible, regardless of whether the existing experiments have considered the same parameter regimes or not. We have presented results where sharp enhancement/suppression is seen, similar to the present experiments, as well as presented results that have interesting features which might not be observed in the current experiments. Thus we do not agree with the characterization that our result as “opposite” to what has been experimentally seen. In addition to this, note that when we find both enhancement or suppression, one of these effects is $\sim 1/5$ the other (see Fig.4), and it's very much possible to miss such weak effects and will likely appear as if only suppression or enhancement is observed. Nevertheless, we have added a sentence to clarify that such results have not been observed yet.

Changes: We have added a few sentences on page 9 : “Our results point to the possibility of observing both suppression and enhancement in the same frequency scale, depending on photon frequency. We note that such behavior (observing both suppression and enhancement) has not yet been observed in present experiments [ref].”

Comment 10. In conclusion, I would only consider to accept this manuscript in Nature Communications after some major modifications which address the issues outlined above. My suggestions to help to improve the manuscript:

Comment 10.1 Motivate the construction of the model in detail and cite previous work as much as possible.

Response. We appreciate this suggestion. We have added text clarifying the choice of our model and we have added additional calculations covering a larger space of parameter regimes..

Changes:

- We have added the text when describing solvent-molecule interactions: “In this work, the essential features of the solvent-molecule interactions are included in the spectral density. In particular, the spectral density can be calculated by molecular dynamics in simulations and often contains sharp peaks [ref]. When this is the case, the spectral density can roughly be grouped into two categories. In the first category are cases where the spectral density has peaks near the molecular vibrational frequencies. The second category comprises cases where the spectral density does not have peaks near the molecular vibrational frequencies. We find that spectral densities with off-resonant peaks exhibit the same cavity modulation of reactivity as spectral densities devoid of peaks, as shown in Supplementary Note 8.”
- We have added the text “In this work we have systematically explored a wide range of parameter regimes associated with cavity modification of chemical reactivity in the single molecule limit. There are three parts of our model system:
 - The molecular sub-system, which is described by a double well potential profile which is widely employed in studying chemical kinetics [ref] and used previously in the theoretical investigation of vibrational polariton chemistry [ref]. We find that, outside of the tunneling dominated regime (see Supplementary Note 3) the rough qualitative features of our results do not sensitively depend on the parameters describing the molecular system, such as the well/barrier frequency, height of the barrier, driving force and or shape of the solute dipole (either linear or nonlinear).
 - The cavity radiation and light-matter interaction term. Here, there are three relevant parameters which characterize this part of our model, namely the cavity photon frequency, light-matter coupling strength, and cavity lifetime. We find that all three parameters play a crucial role in modifying chemical reactivity.
 - The solvent and the molecule-solvent interactions. We find that the spectral density that characterizes the molecule-solvent interactions plays a crucial role in the cavity modification of chemical reactivity.”

Comment 10.2 Include the absorption lineshape of the coupled vibration in order to make the correspondence with the cavity detuning response. The matching of the detuning response to lineshape is the strongest indication experimentalist have currently to show the presence of cavity effects

Response and Changes: We modified Fig. 1 and replaced the absorption spectrum of the coupled system with the absorption spectrum of the bare molecular system. We have provided additional absorption spectra of other systems in the supplementary information.

Comment 10.3 Extend the range of considered coupling strengths. The last results the authors show is a rate suppression if you assume a coupling strength η_c of 0.01 and a large molecule-spectator mode coupling. If you would increase η_c even further, will the observed trend remain? Or will they fall in line with experimental observation? Note that experiments so far have mainly uncovered rate suppressions under relative coupling strengths of at least a few percent (going up to 20 % even when working with water).

Response. It is completely certain that increasing η_c will further decrease the chemical rate. This is because large η_c splits either the molecular vibrations or the spectral density, thereby moving the molecular transition further away from the peaks in the spectral density resulting in reduced interactions between molecules and solvents. We demonstrate this below where we systematically increased the light-matter coupling. As explained and expected, increasing the light-matter coupling increases the suppression. At some η_c , however, this effect will saturate as the interaction between the spectator mode and the molecule becomes minimal. Note that $\eta_c = 0.01$ roughly corresponds to a Rabi-splitting that is $\sim 18\%$ of the resonant photon frequency.

Change:

- We have added these results in **Supplementary Information: Added Fig.S14** and added the text starting with “We perform quantum dynamics simulation in a molecular system coupled to a structured spectral density with the peak of the spectral density originating from a spectator mode ...”

Comment 10.4 Consider the effect of simplified H_{solv} Hamiltonians to show the effect of each single type of individual bath effect they include. This would be comparing just to molecule in the cavity to molecule + spectator mode, molecule + solvent, molecule + solvent + solvent spectator mode and, finally, molecule + spectator mode + solvent + solvent spectator mode.

Response. We believe we have already conducted a thorough investigation on how relevant parts of the bath play a role. Below we comment on each of the scenarios suggested by the reviewer,

1. **molecule in the cavity:** This is a two mode closed quantum system which will exhibit

periodic oscillations, thus there is no chemical rate kinetics in such a system. The closest case to this is that of a molecule in the cavity + solvent with an unstructured spectral density which is presented in **Fig.2 and Fig.3** and has been discussed in detail in the main text (additional results can also be found in the Supplementary Information).

2. **molecule + spectator mode:** Similar to previous example, there exist no well-defined chemical kinetics in such a system.

3. **molecule + solvent + solvent spectator mode:** It is not clear from the reviewers comment what is precisely meant by the “solvent spectator mode”. By “solvent spectator mode” if the reviewer is referring to the secondary solvent coupled to the to the spectator mode then it reduces to the molecule + solvent (unstructured spectral density) case, as this “solvent spectator mode” only interacts with the molecular subsystem through the spectator mode.

4. **molecule + solvent:** We have simulated such a system in detail in **Fig.1**.

5. **molecule + spectator mode + solvent + solvent spectator mode:** Again it is not clear from the reviewer’s comment what is precisely meant by the “solvent spectator mode”. By “solvent spectator mode” if the reviewer is referring to the secondary solvent coupled to the spectator mode then such a system is investigated in detail in **Fig.4 and Fig.5**.

Overall, we have carefully performed our investigations and have clarified how each part of the system plays a role in modifying chemical reactivity. The setups suggested by the reviewer, either do not appear to have a well-defined chemical rate or have been presented and discussed in our work.

No additional changes have been made.

Comment 10.5. Consider an asymmetric potential for the molecule. (Although this might be already beyond the current scope.) A perfectly symmetric potential is actually a very specific case.

Response: We have now presented such results in the supporting information. The main qualitative conclusions obtained in our work remain unchanged. Below we show the modified figure of the such scenario:

Changes: We have presented this result in the Supplementary information (Fig.S4).

Reviewer #3 (Remarks to the Author):

Comment 1. In this article, the authors claim to explain a longstanding open question regarding reaction rates of molecules in a cavity resonant with some vibrational transitions, a branch of research having gained a lot of interest lately after a number of experiments, [1-8] in the article. There have been numerous attempts to explain those findings theoretically, but with varying success, and none which could explain all the effects.

The calculations here are very thorough and the theoretical curves coincide nicely with the experimental ones. This looks very promising. On the other hand, with an ansatz with multiple Hamiltonians and a truly enormous number of adjustable parameters I would be surprised if those curves could not be fitted.

As the paper stands now, it is not clear to me whether this is really an explanation or just a good fitting.

Response: We thank the reviewer for the encouraging words. We assure the reviewer that the results obtained in this work are general for the physical class of systems we consider (we note we operate in the single molecule regime and not the collective regime), and they are not matching experimental observations (where they do) because we have chosen some *post facto* optimal parameter set.

Overall there are three parts to our model:

- One, the molecular sub-system which is described by a double well potential which is widely employed in studying chemical kinetics (J. Chem. Phys. 127, 144503 (2007), J. Chem. Phys. 101, 7500 (1994), J. Phys. Chem. 99, 9, 2777–2781, (1995), Rev. Mod. Phys. 62, 251, 1990) and has been used in theoretical investigation of vibrational polariton chemistry (Nat. Comm. 12, 1315 (2021), Phys. Rev. X 9, 021057 (2019), J. Chem. Phys. 156, 154305 (2022), J. Phys. Chem. Lett., 12, 29, 6974–6982 (2021), J. Phys. Chem. Lett. 13 (28) 6580 (2022), J. Phys. Chem. Lett. 12, 39, 9531–9538 (2021)). We find that, outside of the tunneling dominated regime (see Supplementary Information Fig. S3), the rough qualitative features of our results do not sensitively depend on the parameters describing the molecular system, such as the well/barrier frequency, height of the barrier, driving force and the shape of the dipole (linear/non-linear).

We have performed additional calculations which should convince the reviewer that our results are general:

1. We have simulated model system with three different barriers – 2250 cm^{-1} (main text), 3000 cm^{-1} (Fig.S11) and 4000 cm^{-1} (Fig.S12) – all show similar results.
2. We have simulated symmetric (main text) and asymmetric systems (Fig. S4), both of which show similar results.
3. We have simulated linear (main text) and non-linear dipoles (Fig. S10) which show similar results.

Therefore we find that by scanning a large space of molecular parameter regimes, our conclusions are quite general.

- Second, the cavity radiation and light-matter interactions. There are three relevant parameters, the cavity photon frequency, light-matter coupling strength and cavity lifetime. We have scanned over all three parameters.
 1. We find that a finite cavity lifetime (which inevitably occurs in real experiments) is essential to observe the cavity induced effects. We choose a cavity lifetime that is experimentally reasonable (100 - 1000 fs).
 2. We have simply scanned the cavity coupling strengths. We find that increasing light-matter coupling generally enhances rate modifications by the cavity.
 3. We have simply scanned over the cavity photon frequency range (~500- 2000 cm^{-1}). We have verified that there is no interesting physics outside this range.
- Lastly, the solvent and the molecule-solvent interactions. We find that the spectral density that characterizes the molecule-solvent interactions plays a crucial role in the cavity modification of chemical reactivity.

To understand the role of the spectral density we consider two scenarios that encompass the two typical types of molecule-solvent interactions. The spectral density is generally characterized by sharp peaks originating from molecular vibrations in the solvent as well as broad, low frequency contributions due to translations and rotations of solvent molecules (J. Chem. Phys. 115, 7622 (2001), J. Chem. Phys. 93, 5084 (1990), Chem. Phys. Lett. 292, 431-436, (1998), J. Chem. Phys. 135, 064504 (2011)). The solvent-molecular spectral density can roughly be grouped into two categories.

1. The spectral density of the solvent has peaks near the molecular transitions.
2. The spectral density does not have peaks near the molecular transitions. We find that spectral density with off-resonant peaks exhibit the same cavity modulations as for a spectral density without peaks as shown in Supplementary Note 8.

This is why we consider the two scenarios : an unstructured and a structured spectral density (with peak near the molecular transitions). We have also varied the overall strength of the molecule-solvent interaction.

Thus, while it does seem like there are an enormous number of adjustable parameters, we have, where possible, aimed to either scan over parameter space or use physically relevant parameters. We additionally note that prior theoretical treatments that have employed a classical description of the problem and have considered similar models with scans over similar parameter regimes and have failed to capture the experimentally similar features that we do, thus we hope that the reviewer agrees that these results are not an outcome of just good fitting.

Changes: We have added many new results (some of which were added in response to comments made by the other reviewers as well) in different parameter regimes.

- Added Fig. S11 and Fig. S12 that considers 3000^{-1} and 4000 cm^{-1} barriers, and have added text to describe these results in the Supplementary Information.
- We have added Fig.S10 which describes results with non-linear dipoles.
- We have added Fig.S13 to demonstrate that off-resonant peaks in the spectral density do not contribute to cavity modification of chemical reactivity.
- We added the text (Page 3) “We note that while it appears that there are a large number of adjustable parameters, we have, where possible, aimed to either scan over parameter space or use physically relevant parameters.”
- We have added the text (also in response to previous comments) “In this work we have systematically explored a wide range of parameter regimes associated with cavity modification of chemical reactivity in the single molecule limit. There are three parts of our model system:
 - The molecular sub-system, which is described by a double well potential profile which is widely employed in studying chemical kinetics [ref] and used previously in the theoretical investigation of vibrational polariton chemistry [ref]. We find that, outside of the tunneling dominated regime (see Supplementary Note 3) the rough qualitative features of our results do not sensitively depend on the parameters describing the molecular system, such as the well/barrier frequency, height of the barrier, driving force and or shape of the solute dipole (either linear or nonlinear).
 - The cavity radiation and light-matter interaction term. Here, there are three relevant parameters which characterize this part of our model, namely the cavity photon frequency, light-matter coupling strength, and cavity lifetime. We find that all three parameters play a crucial role in modifying chemical reactivity.
 - The solvent and the molecule-solvent interactions. We find that the spectral density that characterizes the molecule-solvent interactions plays a crucial role in the cavity modification of chemical reactivity. ”
- We added the text on Page 5 starting with (also in response to a previous comment): “In this work, the essential features of the solvent-molecule interactions are included in the spectral density. In particular, the spectral density can be calculated by molecular dynamics in simulations and often contains sharp peaks [ref]. When this is the case, the spectral density can roughly be grouped into two categories. In the first category are cases where the spectral density has peaks near the molecular vibrational frequencies. The second category comprises cases where the spectral density does not have peaks near the molecular vibrational frequencies. We find that spectral densities with off-resonant peaks exhibit the same cavity modulation of reactivity as spectral densities devoid of peaks, as shown in Supplementary Note 8.

This is why in the following simulations we will consider two specific cases for the solvent. In the first, we will consider no coupling between the system and spectator

mode, i.e. $C_Q = 0$. In this case, the spectral density for the solvent is described by the broad, unstructured spectral density $J_U(\omega)$, and as such will be referred to as the “unstructured” environment. In the second, we will consider non-zero coupling between the system and spectator mode. In this case, applying a normal mode transformation to the solvent Hamiltonian, we find that the presence of the spectator mode gives rise to a sharp peak in the solvent spectral density $J_S(\omega)$, as shown in Fig.~\ref{fig1}b). Note that here we use the term “structured” to refer to the sharp structure in the spectral density at specified frequencies which does not imply positional or orientational structure in the environment.”

Comment 2: The authors also emphasize that quantum calculations are necessary. My first impulse on this is “of course” — and some of the prior theory works they authors cite also use a full quantum formalism, thus, I cannot see how the fact that calculations are based on a quantum treatment can be quite as novel and game-changing as the authors seem to claim.

Response: We thank the reviewer for raising this question which allows us to clarify the difference between our work and existing other works. In short, essentially all previous studies have made approximations that either ignore the solvent (such that the reactions aren't in a condensed phase rate regime!) and treat the cavity+molecule quantum mechanically, or do treat a full condensed phase system but do not consider fully quantum degrees of freedom, without which the cavity modifications of chemical rate that we demonstrate are often absent or greatly altered. While we explained this in the introduction “Recent fully quantum dynamical studies which ignore the explicit interactions of the molecule with the solvent degrees of freedom also do not find a resonant structure in the cavity frequency dependence of chemical rate...”, we have now modified the introduction to emphasize that previous work using quantum treatment used approximations or considered systems not relevant for solution phase reactions. We also point out that while the reviewer views the message of the necessity of the full treatment of quantum effects as perhaps unsurprising, we note that the fact that most previous work does not employ a fully quantum mechanical description, and the fact that a different reviewer of this manuscript questions the importance of quantum effects on rate alterations, as evidence that many in the community would not share this viewpoint.

We thus emphasize that our work presents the *first exact open quantum system simulations of vibrational polariton chemistry where chemical rates are calculated exactly*. Crucially, we explicitly model all degrees of freedom, including the solvent interactions, with exact quantum dynamics. Previous work that has made approximations to include the effects of the solvent (using a classical treatment or an approximate quantum treatment) fails to capture the results we demonstrate here. It may seem surprising that one needs a fully quantum treatment of the solvent degrees of freedom, as they often are lower frequency than the remainder of the systems, but as we have argued in the text, vibrational energy relaxation is crucial for the results we observe, and it is known that this cannot be accurately modeled by a mixed quantum-classical approach when the system is high frequency and the bath is low frequency (see: J. S. Bader and B.J. Berne, J. Chem. Phys. **100**, 8359 (1994); S.A. Egorov and B.J. Berne, J. Chem. Phys. **107**, 6050 (1997)).

We thus disagree with the reviewer that previous works have used “a full quantum formalism”. Below we list few works that have used (approximate) quantum treatment to simulate chemical kinetics of vibrational polaritons:

- (Phys. Rev. X 9, 021057, 2019) : Quantum mechanically treats a cavity mode and a reaction coordinate- but *does not include a solvent and does not consider cavity loss* (we find cavity loss to be crucial).
- (J. Phys. Chem. Lett. 12, 39, 9531–9538, 2021) : Uses an *approximate* semi-classical rate theory. This theory makes many assumptions, including: (1) that thermalization occurs nearly instantaneously, an assumption that breaks down completely in the energy diffusion-limited regime. We observe rate enhancement in this regime which is fundamentally beyond the scope of this approximate theory. (2) no cavity loss. (3) approximate product partition function using harmonic approximation for reactant well and barrier region. As a result the theory presented here gives results quite similar to the purely classical theory.
- (J. Chem. Phys. 156 (15), 154305, 2022) : Treats cavity and molecule quantum mechanically. *The solvent is not included and the calculations do not consider cavity loss.*

In summary, while prior work has attempted to use quantum approaches, they either employ simple, approximate quantum corrections to approximate classical rate theories and aren't able to capture the dynamics we expose, or consider small *closed* quantum systems for which the thermal rate constant is not well defined. For such closed systems, the transient dynamics was analyzed in previous work, *but this is not directly relevant for chemical kinetics.*

We have also added new results compared to the classical treatment of the same model, which further demonstrates the importance of a fully quantum mechanical treatment, see our response to Comment 0 of reviewer 1 and Fig. S7, Fig. S8 in the modified draft.

Changes:

- We have modified the introduction on page 1 to emphasize that the previous works were approximate. The modified text reads: “Simple approximate quantum corrections to the GH theory, such as found using quantum corrections to the GH theory, such as found using quantum transition state theory [ref] or zero-point energy corrections to the energy barrier [ref] have been carried out, but these approximate calculations diverge from experimental expectations even more than their fully classical counterparts, showing, for example, an even broader range of alteration of the rate profile than that seen in classical calculations [ref]”
- We have added the sentence on page 2: “We emphasize that compared with previous work [ref] which employs a quantum treatment of the cavity and the molecular degrees of freedom in the absence of a molecular or solvent bath, we explicitly include the dissipation provided by a solvent exactly within our model. As we will discuss, this is crucial for obtaining the sharp, resonant cavity modifications observed here.”

- In the conclusion we add the sentence: “Note that previous studies [ref] that also used a quantum mechanical treatment of cavity and molecule but ignore the solvent fail to capture the features of the cavity modification of chemical kinetics that we expose here. Without a proper account of the dissipation produced by the solvents as well as quantum treatment of cavity loss, a complete picture of how a cavity modifies chemical kinetics cannot be obtained.”

Comment 3: I would strongly suggest to clarify much better what is done - at the moment, any serious information does get lost in “too much stuff.” For example, the manuscript would benefit from a table that summarizes the numerical experiments with respect to the various parameters and that compares the numerical experiments to real ones as well as from better foreshadowing in the text of the main results and logical sequence of the figures.

In more detail:

The summary of the paper from the last couple of paragraphs in the introduction are too vague--could they summarize the main takeaways to look out for in the rest of the text, e.g. the dependence of the reaction rate on whether the reactive mode is coupled to a spectator mode, weaker coupling strengths are necessary to observe changes in reaction rate, finite cavity loss is a crucial ingredient? Overall, the specific and interesting insights are scattered throughout the manuscript, and it would be more convenient for the reader to collect them in one place of the manuscript.

Response: We appreciate this thoughtful suggestion. We have added text and a table summarizing some of the key results presented in this work.

Changes:

- We added this table and accompanying text in the conclusion: “In Table.1, we summarize how coupling to a cavity influences the chemical reactivity for $\{it\}$ some of the models studied in this work.

Cavity	Solvent	Effect
Lossy $q_c - R$	Unstructured Tunneling Dominated	Broad Suppression
Lossy $q_c - R$	Unstructured Energy Diffusion-Limited	Sharp Enhancement
Perfect $q_c - R$	Unstructured Energy Diffusion-Limited	No effect
Lossy $q_c - R$	Structured Energy Diffusion-Limited	Sharp Enhancement and/or Suppression
Lossy $q_c - Q$	Structured Energy Diffusion-Limited	Sharp Enhancement and/or Suppression
Lossy $q_c - R$	Structured Spatial Diffusion-Limited	Sharp Suppression
Lossy $q_c - Q$	Structured Spatial Diffusion-Limited	Sharp Suppression

TABLE I. Summary of cavity modification to chemical reactivity of model systems investigated in this work.

Here $q_c - R$ and $q_c - Q$ signifies coupling the cavity to the reaction coordinate or to the spectator mode, respectively.”

- We have added the following text (Page 11) “Overall, for unstructured spectral densities we observed sharp resonant cavity modification of the chemical rate in the energy diffusion-limited for finite cavity lifetime. For structured spectral densities we find that it is possible to obtain sharp suppression or enhancement of the rate, depending on whether the cavity coupling increases or decreases the solvent-molecule interactions. When some solvent vibration modes have frequencies close to molecular vibrational transitions, the net coupling between the molecule and the solvent is modified by either coupling the cavity to the molecular vibrations or to the solvent mode. Coupling the cavity radiation mode to molecular vibration splits the molecular vibrational states, thereby either moving them closer or further away from the solvent modes. A similar situation occurs when coupling the cavity radiation mode to the solvent modes. Whether this leads to enhancement or suppression (or both) depends on whether solvent-molecule coupling strength places the system in the energy or spatial diffusion-limited regimes, and on how molecule solvent interactions are modified. Note that because such effects originate from the requirement of state splitting, something that only takes place near resonance, we observe sharp photon frequency dependence of the cavity modified rate. ”

Comment 4: Along these lines, it would be helpful to summarize earlier in the paper what studies are done and why. Currently, the structure of the paper is such that experiments are done with parameter X turned off with parameter Y swept and parameter Z sometimes on, and it's difficult to keep up with why one calculation was done and not some other one and how calculation A led to calculation B.

Response: We have significantly modified our text (in response to previous comments by the referees as well) to explain why we have chosen to study these model systems.

Changes:

- We have added the text (also in response to previous comments) :
“In this work we have systematically explored a wide range of parameter regimes associated with cavity modification of chemical reactivity in the single molecule limit. There are three parts of our model system:
 - The molecular sub-system, which is described by a double well potential profile which is widely employed in studying chemical kinetics [ref] and used previously in the theoretical investigation of vibrational polariton chemistry [ref]. We find that, outside of the tunneling dominated regime (see Supplementary Note 3) the rough qualitative features of our results do not sensitively depend on the parameters describing the molecular system, such as the well/barrier frequency, height of the barrier, driving force and or shape of the solute dipole (either linear or nonlinear).
 - The cavity radiation and light-matter interaction term. Here, there are three relevant parameters which characterize this part of our model, namely the cavity photon frequency, light-matter coupling strength, and cavity lifetime. We find that all three parameters play a crucial role in modifying chemical reactivity.

The solvent and the molecule-solvent interactions. We find that the spectral density that characterizes the molecule-solvent interactions plays a crucial role in the cavity modification of chemical reactivity.”

- We have added the text (also in response to previous comments) :

“In this work, the essential features of the solvent-molecule interactions are included in the spectral density. In particular, the spectral density can be calculated by molecular dynamics in simulations and often contains sharp peaks [ref]. When this is the case, the spectral density can roughly be grouped into two categories. In the first category are cases where the spectral density has peaks near the molecular vibrational frequencies. The second category comprises cases where the spectral density does not have peaks near the molecular vibrational frequencies. We find that spectral densities with off-resonant peaks exhibit the same cavity modulation of reactivity as spectral densities devoid of peaks, as shown in Supplementary Note 8.

This is why in the following simulations we will consider two specific cases for the solvent. In the first, we will consider no coupling between the system and spectator mode, i.e. $C_Q = 0$. In this case, the spectral density for the solvent is described by the broad, unstructured spectral density $J_U(\omega)$, and as such will be referred to as the “unstructured” environment. In the second, we will consider non-zero coupling between the system and spectator mode. In this case, applying a normal mode transformation to the solvent Hamiltonian, we find that the presence of the spectator mode gives rise to a sharp peak in the solvent spectral density $J_S(\omega)$, as shown in Fig.~\ref{fig1}b). Note that here we use the term “structured” to refer to the sharp structure in the spectral density at specified frequencies which does not imply positional or orientational structure in the environment.”

Comment 5: The variables in the expressions for the spectra densities should be labelled and described (the various Λ , Γ , Ω , η)

Response: We appreciate the suggestion. We have described these terms.

Changes: Added the following texts

- “Here Λ_s corresponds to the solvent reorganization energy, Ω is the characteristic frequency that determines the peak of the spectral density and $\eta_s = 2\Lambda_s$ is the friction constant.”
- “Here, λ_s and γ are the reorganization energy and the characteristic frequency of this secondary solvent bath.”
- “Here, λ_s and γ are the reorganization energy and the characteristic frequency of this secondary solvent bath.”

Comment 6: Since Fig 1 focuses on different limits of H_{solV} , it may be useful to describe structured vs. unstructured environments earlier

Fig 1 is a little confusing at first because the article is presumably about the quantum dynamics of vibrational polariton chemistry, but this is not described until Fig 2, and there is no visual direct comparison between cavity-free and cavity reactions. Confusingly, the authors focus first on the difference between structured and unstructured solvents, which is generally interesting but seems off-topic in this article. Instead, perhaps a more relevant and pressing discussion would be the difference between structured and unstructured molecule-solvent coupling inside of a cavity.

Response: We emphasize that the mechanistic principle behind the cavity modification of chemical reactivity, as revealed in our simulations, is related to how the solvent modifies the chemical rate. As was explained in our main text, “the extent and the nature (enhancement or suppression) of the cavity modification to the reaction rate depends sensitively on (a) the details of the molecular system, such as the potential energy surface and the vibrational eigenspectrum, (b) the details of the solvent, as encoded in the spectral density and solvent friction ... the cavity modification of chemical rates can largely be rationalized by considering how the molecular vibrational states are altered by hybridization with the cavity photon states (forming so-called vibrational polaritons) to effectively increase or decrease the interaction of the molecule with its environment.”

Therefore, it is essential that we introduce the role of solvent-molecule coupling in chemical kinetics before we show how the presence of the cavity modifies chemical reactivity. This leads us to believe that the current arrangement of figures is optimal.

No change is made.

Comment 7: There is generally a lack of mechanistic explanations, especially why quantum calculations give such different results than classical ones. For instance, In Fig 2d, why is the rate constant peaked at the lower polariton energy, and in Fig 2e, why are the resonant frequencies different for the classical vs. quantum cases? In the "Summary" section, it is stated that these differences are due to "quantum light-matter hybridization" and elsewhere, some resonance shifts are due to "renormalization." Both light-matter hybridization and renormalization can also be described classically--what is uniquely "quantum" about this phenomenon that leads to qualitatively different reactivity in a cavity?

Response: We have added an explanation as to why the classical theories fail to capture the results obtained in our quantum dynamics simulations (also see our response to comment.0 of reviewer 1). The fundamental reason why a classical treatment breaks down is because it cannot accurately capture vibrational energy transfer beyond the Golden Rule limit and/or in the presence anharmonicity e.g. the double-well potential along the reaction coordinate. The lessons learned from the study of vibrational energy relaxation, which is a crucial step in the polaritonic modification of chemical reaction rates, are critical for understanding when we can expect a breakdown of the classical treatment for reaction rates inside a cavity. We expect that there are two potentially simultaneously operative regimes where classical calculations will be in error:

- When the Golden Rule-level of theory for rates cannot be applied. As shown by Bader and Berne (J.S. Bader and B.J. Berne, J. Chem. Phys. 100, 8359 (1994)) the *Golden Rule-level* theory of vibrational relaxation rates for a purely harmonic, purely classical system is identical to that of a purely harmonic quantum system. However this agreement will break down sharply when one needs to go *beyond* the Golden Rule to capture the relaxation rates. We expect that this is the case in our system as the rate is governed by resonances, obviating low-order perturbation theory.
- As shown by Egorov and Berne (S.A. Egorov and B.J. Berne, J. Chem. Phys. 107, 6050 (1997)) when a system is non-linear, then purely classical theories of vibrational relaxation, while potentially preferable to mixed quantum-classical ones, are not quantitatively accurate even in the Golden Rule limit, especially for high frequency system modes, regardless of the bath frequencies. In the case studied by Egorov and Berne the non-linearity comes from the form of the system-bath coupling, while in our case it comes from the form of the potential energy curve along the reaction coordinate. It is simple to show that these cases will behave similarly by writing our Hamiltonian in the basis of vibrational eigenfunction of the system. In this basis the system-bath term will couple all eigenstates, and not just ones that differ by one vibrational quantum, just as in the model of Egorov and Berne.

We believe both aspects outlined above are in play in the systems we study (we will give more evidence for a breakdown of classical mechanics in Fig.R1-2 in response to comment.0 of reviewer 1.

In particular, it should be noted that cavity radiation *strongly and resonantly* couples to the molecule. This fact renders perturbative, Golden Rule treatments, where classical mechanics can often give sensible answers, unreliable. Further, the non-linear nature of the reaction coordinate potential energy profile can also render a classical theory inaccurate. As an illustration of the former point, consider the Fig.R3 where we compare IR and cavity modified chemical rate computed quantum mechanically (Fig.R3a) and classically (Fig.R3b). The thickness of the classical IR is very close to the quantum IR spectra while the differences in the rates are much more drastic. This difference here is due to the fact that the IR spectrum, is accurately described within the Golden Rule formalism, while the chemical kinetics inside the cavity, where resonances are important, is not.

We believe the reviewer has misunderstood the renormalization that we discuss. Here, we are referring to the renormalization of the *tunneling amplitude* (namely Δ is multiplied by Franck-Condon factors) which has no relationship to resonance shifts. This cannot be described classically. We have clarified this in the text.

Changes:

- We added the text after the word renormalization (Page 10) : “rescaling of \$\Delta \rightarrow \Delta \cdot \langle \exp[i \eta_c \sqrt{2\omega_c} \hat{p}_c / \mu] \rangle\$ due to the difference in permanent dipoles”
- We have added discussion of how the classical description breaks down : “In Supplementary Note 4, we compare the classical rate profile to the IR spectrum obtained using classical methods and observe discrepancies between the peak locations of the two profiles. Interestingly, the width of the classical absorption profile is as sharp as its quantum counterpart shown in Fig.~\ref{fig2}d (also see Fig.S6) in

contrast to the width of the chemical rate profile obtained classically. This difference is due to the fact that the IR spectrum, which depends on molecule-solvent interactions, is accurately described within the Golden Rule formalism, while the chemical kinetics inside the cavity, where resonance effects are important, is not.”

- We have added the text thoroughly explaining the breakdown of classical mechanics starting with “The lessons from the study of vibrational energy relaxation, which is a crucial step in the polaritonic modification of chemical reaction rates, are critical for understanding when we can expect a breakdown of the classical treatment for reaction rates inside a cavity. We expect that there are two potentially simultaneously operative regimes where classical calculations will be in error:
 - When a Golden Rule-level of theory for rates cannot be applied. As shown by Bader and Berne [ref] the Golden Rule vibrational relaxation rate for a purely harmonic, purely classical system is identical to that of a purely harmonic quantum system. However this agreement will break down sharply when one needs to go beyond the Golden Rule to capture the relaxation rates. We expect that this is the case in our system as the rate is governed by resonances, obviating low-order perturbation theory.
 - As shown by Egorov and Berne [ref] when a system is nonlinear, then purely classical theories of vibrational relaxation, while potentially preferable to mixed quantum-classical ones, are not quantitatively accurate even in the Golden Rule limit, especially for high frequency system modes, regardless of the bath frequencies. In the case studied by Egorov and Berne the nonlinearity comes from the form of the system-bath coupling, while in our case it comes from the form of the potential energy curve along the reaction coordinate. It is simple to show that these cases will behave similarly by writing our Hamiltonian in the basis of vibrational eigenfunction of the system. In this basis the system-bath term will couple all eigenstates, and not just ones that differ by one vibrational quantum, just as in the model of Egorov and Berne [ref].”

Comment 8: There are many variables in this complex model. Could the authors summarize (a) which parameter regimes lead to modifications of chemical reactivity and (b) which parameter regimes real experiments correspond to? As of now, it is challenging to understand whether this theory applies at all to experiments completed thus far, which is fine in my opinion but just needs to be evident

Response: We have clarified the role of various parameters used in our work and added discussion on the meaning of the model and how the model is constructed. We have also added a table as suggested by the reviewer. These changes taken together should be sufficient to obtain a clearer picture of what results we have obtained.

Comment 9: In summary, in its present form, I don't think this manuscript is suitable yet for publication.

Response: We hope that our modifications, clarifications, and additional results will convince the reviewer to recommend our work for publication, and we thank him/her for their detailed and helpful review.

REVIEWER COMMENTS

Reviewer #1 (Remarks to the Author):

I have carefully read the answers to my and to the other Referee's comments, and the modified manuscript. In my opinion, the authors have done a tremendous job at considering and answering all points, and where necessary have performed new simulations, which now further strengthen their arguments.

In general, I am still of the opinion that the full quantum treatment does not change the fundamental mechanism of operation of the cavity, which is to assist or hinder the energy redistribution pathways of the molecule, meaning that the effects are, to a large extent, dynamical and not related to modifications of the equilibrium state energies and barrier height or its curvature.

This does not diminish this work's high level of insight, which I now understand better. The fact that experimentally the cavity modifications of the rates are within an order of magnitude or less, which is within the accuracy expected for classical theories, calls for a fully quantum treatment of a general enough model that, at least for the strong coupling case of one single molecule, explains the key features of the mechanism without further approximation.

The point for the fully quantum simulations is now much more clearly stated in the paper and the new text on page 11 has a very high pedagogical value for those of us less familiar with condensed phase simulations.

Summarising, I think that this is a very significant contribution to the overall theoretical advancement of the field, and I strongly recommend its publication in Nat. Commun.

Reviewer #2 (Remarks to the Author):

Overall, I am glad to see to the author addressing the comments and questions of myself and the other two reviewers in such length. Of particular importance are the language clarifications (see Comment 4). The field of vibrational strong coupling (VSC) is mix of people with different backgrounds, which makes language a hurdle to overcome.

As a result, the manuscript is now significantly improved and now constitutes a nice contribution to the field. It is an interesting step in our search for a theory for chemical reactions under VSC advancing some new ideas (role dissipation & solvent, the possibility of having both rate enhancements and suppressions) and some numerically exact computational results. I now see that there is some correspondence to available experimental results and this is at least better than for many other models proposed to this date.

There are two major trade-offs in order to have exact numerical solubility, which are ignoring anharmonicity and collectivity. As the authors point out, the former has been done for traditional theories of chemical kinetics (e.g. transition state theory) and can therefore be acceptable/useful. However, not taking collectivity into account may very well make this work irrelevant or invalid quite soon. The field has converged around the idea that collectivity is essential and is now actively working to clarify this. So, it might well be that in the near future a model is published taking collectivity into full consideration and superseding the model presented here. It needs to be noted to the present work is a rather nice setup for the authors to go into more advanced modeling and discern collective effects based on comparison with this work.

All this being said, there is still one important point with which I have a major issue. This is to such an extent that I cannot accept this manuscript unless it is fully addressed.

The authors insist on stating that single molecule VSC experiments are possible, something I and many other experimentalists wholeheartedly think is not. (see Comment 2 in rebuttal letter) The authors cite two recent papers to support their claim.

The first, J. Phys. Chem. Lett. 2021, 12, 3171–3175, deals with VSC of the C=O vibration of PMMA layer by a plasmonic nanogap. This is overall nice work and I thank the authors to bring it to my attention. However, it does not constitute vibrational strong coupling of a single dipole/molecule. Inside the nanogap, which is typically $1.4\ \mu\text{m} \times 1.4\ \mu\text{m} \times 100\ \text{nm}$, there are still many, many C=O moieties because the gap is completely filled by a PMMA layer. My estimate is that there are on the order of 10^9 C=O groups (using PMMA density = 1.18 g/ml, monomer unit molecular weight = 104 g/mol and gap dimension stated above). From an cQED viewpoint, each of these groups constitutes an individual oscillator dipole which is coupled to the cavity.

The second, J. Phys. Chem. Lett. 2022 13 (41), 9673-9678, is a theoretical paper which predicts single molecule VSC from plasmonic nanogap cavities. Crucially, this paper does not mention the value used for the molecular dipole strength, which is a serious flaw of both the authors and the reviewers of this particular paper. Many theoretical groups in the field use unrealistic large values in order to model cavity with only a single vibration. This makes me suspect that this is also the case here. As a result, you cannot trust this paper. Also note that in plasmonic cavities, you have Coulombic coupling instead of dipolar coupling. The Pauli-Fierz Hamiltonian, used by the authors, assumes the latter.

In other words, the authors cannot use these papers to substantiate their claim about the possibility of single molecule VSC experiments.

Furthermore, the authors claim that “while the oscillator strength is at least 2 orders of magnitude lower in the IR regime, the couplings/Rabi-splittings required for modifying chemical reactivity in the IR regime are also smaller by a few orders of magnitude”. This is too simplistic and not true. In fact, the Rabi splittings in VSC are typically only about approx. 1 order magnitude lower than for electronic strong coupling (30-10 meV vs 500-100 meV). Apart from the lower oscillator strength you also have a higher cavity mode volume, as this scales with the wavelength of the transition you couple to the cavity.

As a consequence of all the above, I cannot approve this paper unless the possibility of single molecule VSC experiments is not mentioned anymore.

Please also note that the possibility of single molecule VSC experiments is not needed at all to make this paper interesting. Many groups have published models of VSC chemistry using a based on single molecule models. The most important things to take away from these are the overall trends, e.g. detuning or Rabi splitting dependence, and included features, like e.g. dissipation, which influence the dynamics. Lastly, chemistry has always and continues to be done on large numbers of molecules. Single molecule experiments are a relative recent development and are interesting mainly for answering some certain particular questions, mainly in heterogeneous, out-of-equilibrium systems. These are systems where ergodicity doesn't hold and/or you want to know the ensemble distributions instead of ensemble averages. The people doing VSC chemistry experiments are mainly chemists by training and adhere to this mind set.

Reviewer #3 (Remarks to the Author):

The authors have extensively explained their reasoning to all three reviewers. Many of the first two reviewers' questions were very specific to their own expertise, thus, I leave it to them to assess whether their concerns were taken into account fully.

My concerns were mostly with the fact, that in my opinion, much of the valuable information was buried too deep into the manuscript to be comprehensible. This had two flavors: First,

that the simulations were done with so many free parameters that could be adjusted that one could fit pretty much any curve. While there are still the same number of parameters, they have been, in the new version, put much better into context and comparison with experiment. While I still think that there is a danger to "mis-compare" because of too many parameters, I do understand from the authors that there are that many degrees of freedom and a simulation with considerably less parameters is probably indeed futile. The second flavor of "too much" was that relevant information was dug too deeply into the text, and I think the authors have now taken care of this to a reasonable degree.

I still hope that somebody at some point comes to a somewhat more elegant solution that can be done with much less parameter-adjusting. But this might very well be elusive. Thus, my concerns have been taken into account, and I would agree to publication in Nat. Comm. if my reviewer colleagues come to the same conclusion.

Reviewer #1:

Comment: I have carefully read the answers to my and to the other Referee's comments, and the modified manuscript. In my opinion, the authors have done a tremendous job at considering and answering all points, and where necessary have performed new simulations, which now further strengthen their arguments.

In general, I am still of the opinion that the full quantum treatment does not change the fundamental mechanism of operation of the cavity, which is to assist or hinder the energy redistribution pathways of the molecule, meaning that the effects are, to a large extent, dynamical and not related to modifications of the equilibrium state energies and barrier height or its curvature.

This does not diminish this work's high level of insight, which I now understand better. The fact that experimentally the cavity modifications of the rates are within an order of magnitude or less, which is within the accuracy expected for classical theories, calls for a fully quantum treatment of a general enough model that, at least for the strong coupling case of one single molecule, explains the key features of the mechanism without further approximation.

The point for the fully quantum simulations is now much more clearly stated in the paper and the new text on page 11 has a very high pedagogical value for those of us less familiar with condensed phase simulations.

Summarising, I think that this is a very significant contribution to the overall theoretical advancement of the field, and I strongly recommend its publication in Nat. Commun.

Response: We thank the reviewer for the thorough review and for recommending our work for publication.

Reviewer #2:

Comment: Overall, I am glad to see to the author addressing the comments and questions of myself and the other two reviewers in such length. Of particular importance are the language clarifications (see Comment 4). The field of vibrational strong coupling (VSC) is mix of people with different backgrounds, which makes language a hurdle to overcome.

As a result, the manuscript is now significantly improved and now constitutes a nice contribution to the field. It is an interesting step in our search for a theory for chemical reactions under VSC advancing some new ideas (role dissipation & solvent, the possibility of having both rate enhancements and suppressions) and some numerically exact computational results. I now see that there is some correspondence to available experimental results and this is at least better than for many other models proposed to this date.

There are two major trade-offs in order to have exact numerical solubility, which are ignoring anharmonicity and collectivity. As the authors point out, the former has been done for traditional theories of chemical kinetics (e.g. transition state theory) and can therefore be acceptable/useful. However, not taking collectivity into account may very well make this work irrelevant or invalid quite soon. The field has converged around the idea that collectivity is essential and is now actively working to clarify this. So, it might well be that in the near future a model is published taking collectivity into full consideration and superseding the model presented here. It needs to be noted to the present work is a rather nice setup for the authors to go into more advanced modeling and discern collective effects based on comparison with this work.

Response: We fully agree with the reviewer and we also share the view that our present work lays the groundwork for more advanced modelling of the cavity modification to chemical reactivity that includes the collective effects. Overall, we thank the reviewer for the thoughtful comments, and we are glad to learn that the reviewer found our manuscript significantly improved.

Comment: All this being said, there is still one important point with which I have a major issue. This is to such an extent that I cannot accept this manuscript unless it is fully addressed.

The authors insist on stating that single molecule VSC experiments are possible, something I and many other experimentalists wholeheartedly think is not. (see Comment 2 in rebuttal letter) The authors cite two recent papers to support their claim.

The first, J. Phys. Chem. Lett. 2021, 12, 3171–3175, deals with VSC of the C=O vibration of PMMA layer by a plasmonic nanogap. This is overall nice work and I thank the authors to bring it to my attention. However, it does not constitute vibrational strong coupling of a single dipole/molecule. Inside the nanogap, which is typically $1.4 \mu\text{m} \times 1.4 \mu\text{m} \times 100 \text{nm}$, there are still many, many C=O moieties because the gap is completely filled by a PMMA layer. My estimate is that there are on the order of 10^9 C=O groups (using PMMA density = 1.18 g/ml, monomer unit molecular weight = 104 g/mol and gap dimension stated above). From a cQED viewpoint, each of these groups constitutes an individual oscillator dipole which is coupled to the cavity.

The second, J. Phys. Chem. Lett. 2022 13 (41), 9673-9678, is a theoretical paper which predicts single molecule VSC from plasmonic nanogap cavities. Crucially, this paper does not mention the value used for the molecular dipole strength, which is a serious flaw of both the authors and the reviewers of this particular paper. Many theoretical groups in the field use unrealistic large values in order to model cavity with only a single vibration. This makes me suspect that this is also the case here. As a result, you cannot trust this paper. Also note that in plasmonic cavities, you have Coulombic coupling instead of dipolar coupling. The Pauli-Fierz Hamiltonian, used by the authors, assumes the latter.

In other words, the authors cannot use these papers to substantiate their claim about the possibility of single molecule VSC experiments.

Furthermore, the authors claim that “while the oscillator strength is at least 2 orders of magnitude lower in the IR regime, the couplings/Rabi-splittings required for modifying chemical reactivity in the IR regime are also smaller by a few orders of magnitude”. This is too simplistic and not true. In fact, the Rabi splittings in VSC are typically only about approx. 1 order magnitude lower than for electronic strong coupling (30-10 meV vs 500-100 meV). Apart from the lower oscillator strength you also have a higher cavity mode volume, as this scales with the wavelength of the transition you couple to the cavity.

As a consequence of all the above, I cannot approve this paper unless the possibility of single molecule VSC experiments is not mentioned anymore.

Please also note that the possibility of single molecule VSC experiments is not needed at all to make this paper interesting. Many groups have published models of VSC chemistry using a based on single molecule models. The most important things to take away from these are the overall trends, e.g. detuning or Rabi splitting dependence, and included features, like e.g. dissipation, which influence the dynamics. Lastly, chemistry has always and continues to be done on large numbers of molecules. Single molecule experiments are a relative recent development and are interesting mainly for answering some certain particular questions, mainly in heterogeneous, out-of-equilibrium systems. These are systems where ergodicity doesn't hold and/or you want to know the ensemble distributions instead of ensemble averages. The people doing VSC chemistry experiments are mainly chemists by training and adhere to this mind set.

Response: We thank the reviewer for the insightful comment. Following the reviewer's suggestion, we removed the sentences where we claimed that a single molecule experiment is possible.

Changes: We removed the following sentences (line numbers from previous revision):

- Lines 152-156: “... and further that single molecule studies are possible in principle”

- Lines 161-164: ~~"The parameters used in this work are well within the range of theoretically estimated ..."~~
- Lines 432-439: ~~"This difference is significant, as it implies that the effects we observe would"~~
- Lines 882-886: ~~"In fact, the magnitude of the effects we see are surprisingly large given that we employ realistically small couplings ..."~~

Reviewer #3

Comment: The authors have extensively explained their reasoning to all three reviewers. Many of the first two reviewers' questions were very specific to their own expertise, thus, I leave it to them to assess whether their concerns were taken into account fully.

My concerns were mostly with the fact, that in my opinion, much of the valuable information was buried too deep into the manuscript to be comprehensible. This had two flavors: First, that the simulations were done with so many free parameters that could be adjusted that one could fit pretty much any curve. While there are still the same number of parameters, they have been, in the new version, put much better into context and comparison with experiment. While I still think that there is a danger to "mis-compare" because of too many parameters, I do understand from the authors that there are that many degrees of freedom and a simulation with considerably less parameters is probably indeed futile. The second flavor of "too much" was that relevant information was dug too deeply into the text, and I think the authors have now taken care of this to a reasonable degree.

I still hope that somebody at some point comes to a somewhat more elegant solution that can be done with much less parameter-adjusting. But this might very well be elusive. Thus, my concerns have been taken into account, and I would agree to publication in Nat. Comm. if my reviewer colleagues come to the same conclusion.

Response: We thank the reviewer for the insightful comments and for agreeing to recommend our work.